# 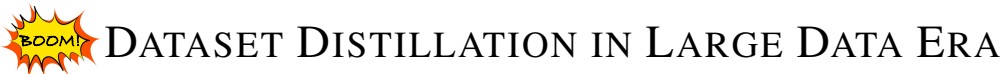 DATASET DISTILLATION IN LARGE DATA ERA

## ABSTRACT

Dataset distillation aims to generate a smaller but representative subset from a large dataset, which allows a model to be trained efficiently, meanwhile evaluating on the original testing data distribution to achieve decent performance. Many prior works have aimed to align with diverse aspects of the original datasets, such as matching the training weight trajectories, gradient, feature/BatchNorm distributions, etc. In this work, we show how to distill various large-scale datasets such as full ImageNet-1K/21K under a conventional input resolution of 224×224 to achieve the best accuracy over all previous approaches, including SRe²L, TESLA and MTT. To achieve this, we introduce a simple yet effective **C**urriculum **D**ata **A**ugmentation (CDA) during data synthesis that obtains the accuracy on large-scale ImageNet-1K and 21K with 63.2% under IPC (Images Per Class) 50 and 36.1% under IPC 20, respectively. Finally, we show that, by integrating all our enhancements together, the proposed model beats the current state-of-the-art by more than 4% top-1 accuracy on ImageNet-1K and for the first time, reduces the gap to its full-data training counterpart to less than absolute 15%. Moreover, this work represents the inaugural success in dataset distillation on larger-scale ImageNet-21K under the standard 224×224 resolution. Our distilled ImageNet-21K dataset of 20 IPC, 2K recovery budget are available anonymously at link.

## 1 INTRODUCTION

Dataset distillation (Wang et al., 2018) has attracted considerable attention across various fields of computer vision (Cazenavette et al., 2022b; Cui et al., 2023; Yin et al., 2023) and natural language processing (Sucholutsky & Schonlau, 2021; Maekawa et al., 2023). This task aims to optimizing the process of condensing massive datasets into a smaller, yet representative subset, preserving the essential features and characteristics that would allow a model to learn from scratch as effectively from the distilled dataset as it would from the original large dataset. As the scale of data and mod-

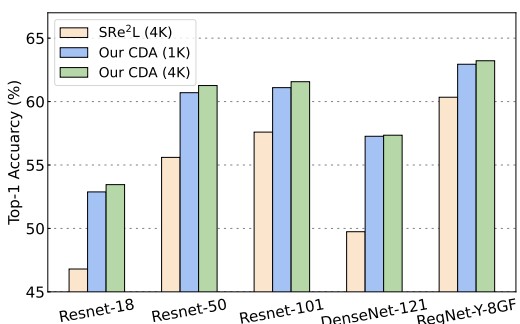

Figure 1: ImageNet-1K comparison with SRe²L.

els continue to grow, this *dataset distillation* concept becomes even more critical in the large data era, where datasets are often so voluminous that they pose storage, computational, and processing challenges. Generally, dataset distillation can level the playing field, allowing researchers with limited computation and storage resources to participate in state-of-the-art foundational model training and application development, such as affordable ChatGPT (Brown et al., 2020; OpenAI, 2023) and Stable Diffusion (Rombach et al., 2022), in the current large data and large model regime. Moreover, by working with distilled datasets, there is potential to alleviate some data privacy concerns, as raw, personally identifiable data points might be excluded from the distilled version.

Recently, there has been a significant trend in adopting large models and large data across various research and application areas. Yet, many prior dataset distillation methods (Wang et al., 2018; Zhao et al., 2020; Zhou et al., 2022; Cazenavette et al., 2022a; Kim et al., 2022a; Cui et al., 2023) predominantly target datasets like CIFAR, Tiny-ImageNet and downsampled ImageNet-1K, find-

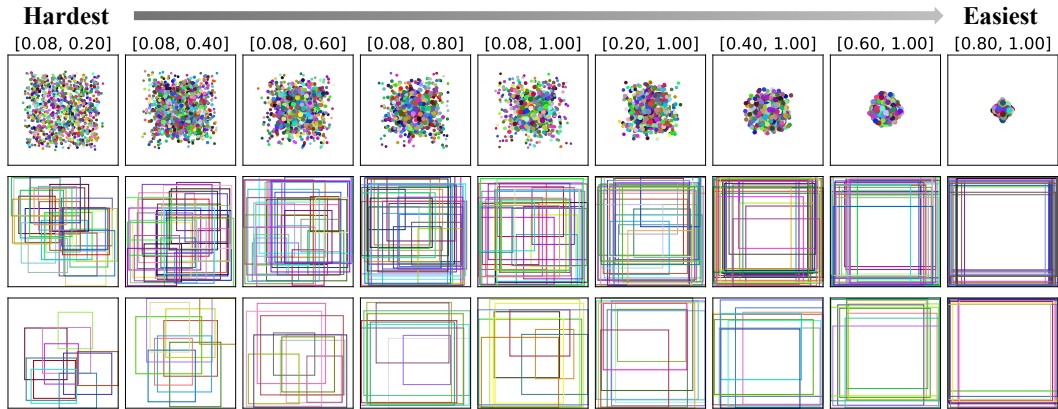

Figure 2: Illustration of crop distribution from different lower and upper bounds in *RandomResizedCrop*. The first row is the central points of bounding boxes from different sampling scale hyperparameters. The second and last rows correspond to 30 and 10 bounding boxes of the crop distributions. In each row, from left to right, the difficulty of crop distribution is decreasing.

ing it challenging to scale their frameworks for larger datasets, such as full ImageNet-1K (Deng et al., 2009). This suggests that these approaches have not fully evolved in line with contemporary advancements and dominant methodologies.

In this study, we extend our focus even beyond the ImageNet-1K dataset, venturing into the uncharted territories of the full ImageNet-21K (Deng et al., 2009; Ridnik et al., 2021) at a conventional resolution of 224×224. This marks a pioneering effort in handling such a vast dataset for dataset distillation task. Our approach harnesses a straightforward yet effective curriculum learning framework. We meticulously address each aspect and craft a robust strategy to effectively train on the complete ImageNet-21K, ensuring comprehensive knowledge is captured. Specifically, following a prior study (Yin et al., 2023), our approach initially trains a model to encapsulate knowledge from the original datasets within its dense parameters. However, we introduce a novel training recipe that surpasses the results of Ridnik et al. (2021) on ImageNet-21K. During the data recovery/synthesis phase, we employ a strategic learning scheme where partial image crops are sequentially updated based on the difficulty of regions: transitioning either from simple to difficult, or vice versa. This progression is modulated by adjusting the lower and upper bounds of the *RandomReiszedCrop* data augmentation throughout varying training iterations. Remarkably, we observe that this straightforward learning approach substantially improves the quality of synthesized data. In this paper, we delve into three learning paradigms for data synthesis linked to the curriculum learning framework. The first is the standard curriculum learning, followed by its alternative approach, reverse curriculum learning. Lastly, we also consider the basic and previously employed method of constant learning.

**Curriculum Learning (CL) and Reverse Curriculum Learning (RCL).** In traditional learning scenarios, a model is trained on a dataset without paying specific attention to the order or difficulty of the data samples. However, humans and animals often learn more effectively when they are first introduced to simpler concepts and then gradually exposed to more complex ones. This educational concept, known as "curriculum learning" (Bengio et al., 2009), has been borrowed and applied to the domain of machine learning (Jiang et al., 2018; Ren et al., 2018; Hacohen & Weinshall, 2019). Unlike curriculum learning that starts with the easiest tasks, reverse curriculum learning begins with the most challenging ones. This learning strategy is a *backtracking* method that if a model struggles with a particular difficult task, it is allowed to fallback to simpler versions of the task.

**How to Change the Difficulties of Training Samples?** *RandomResizedCrop* randomly crops the image to a certain area and then resizes it back to the pre-defined size, ensuring that the model is exposed to different regions and scales of the original image during training. As illustrated in Figure 2, the difficulty level of the cropped region can be controlled by specifying the lower and upper bounds for the area ratio of the crop. This can be used to ensure that certain portions of the image (small details or larger context) are present in the cropped region. If we aim to make the learning process more challenging, reduce the minimum crop ratio. This way, the model will often see only small portions of the image and will have to learn from those limited contexts. If we want the model to see a larger context more frequently, increase the minimum crop ratio. In this paper,

we perform a comprehensive study on how the gradual difficulty changes by sampling strategy influence the optimization of data generation and the quality of synthetic data for dataset distillation task. Our proposed **C**urriculum **D**ata **A**ugmentation (CDA) is a heuristic and intuitive approach to simulate a curriculum learning procedure. Moreover, it is highly effective on large-scale datasets like ImageNet-1K and 21K, achieving state-of-the-art performance on dataset distillation.

**Our Motivation and Intuition.** Both SRe$^2$L (Yin et al., 2023) and our proposed CDA method utilize local batch mean and variance statistics to match the global statistics of the entire original dataset, synthesizing data by applying gradient updates directly to the image. The impact of such a strategy is that the initial few iterations set the stage for the global structure of the ultimately generated image. However, SRe$^2$L does not capitalize on this characteristic. In contrast, CDA efficiently exploits it by initially employing large crops to capture a more accurate outline of the object. As the process progresses, CDA incrementally reduces the crop size to enhance the finer, local details of the object, significantly elevating the quality of the synthesized data.

In this work, we conduct extensive experiments on the Tiny-ImageNet, ImageNet-1K, and ImageNet-21K datasets. Employing a resolution of 224×224 and IPC 50 on ImageNet-1K, the proposed approach attains an impressive accuracy of 63.2%, surpassing all prior state-of-the-art methods by substantial margins. As illustrated in Figure 1, our proposed CDA outperforms SRe$^2$L by 4~6% across different architectures under 50 IPC, on both 1K and 4K recovery budgets. When tested on ImageNet-21K with IPC 20, our method achieves an top-1 accuracy of 35.3%, which is closely competitive, exhibiting only a minimal gap compared to the model pretrained with full data, at 44.5%, while using 50× fewer training samples.

## 2 APPROACH

### 2.1 DATASET DISTILLATION

The goal of dataset distillation is to derive a concise synthetic dataset that maintains a significant proportion of the information contained in the original, much larger dataset. Suppose there is a large labeled dataset $\mathcal{D}_o = \left\{ (\boldsymbol{x}_1, \boldsymbol{y}_1), \ldots, (\boldsymbol{x}_{|\mathcal{D}_o|}, \boldsymbol{y}_{|\mathcal{D}_o|}) \right\}$, our target is to formulate a compact distilled dataset, represented as $\mathcal{D}_d = \left\{ (\boldsymbol{x}'_1, \boldsymbol{y}'_1), \ldots, \left( \boldsymbol{x}'_{|\mathcal{D}_d|}, \boldsymbol{y}'_{|\mathcal{D}_d|} \right) \right\}$, where $\boldsymbol{y}'$ is the soft label coresponding to synthetic data $\boldsymbol{x}'$, and $|\mathcal{D}_d| \ll |\mathcal{D}_o|$, preserving the essential information from the original dataset $\mathcal{D}_o$. The learning objective based on this distilled synthetic dataset is:

$$\boldsymbol{\theta}_{\mathcal{D}_d} = \arg\min_{\boldsymbol{\theta}} \mathcal{L}_{\mathcal{D}_d}(\boldsymbol{\theta}) \tag{1}$$

$$\mathcal{L}_{\mathcal{D}_d}(\boldsymbol{\theta}) = \mathbb{E}_{(\boldsymbol{x}', \boldsymbol{y}') \in \mathcal{D}_d} \Big[ \ell(\phi_{\boldsymbol{\theta}_{\mathcal{D}_d}}(\boldsymbol{x}'), \boldsymbol{y}') \Big] \tag{2}$$

where $\ell$ is the regular loss function such as the soft cross-entropy, and $\phi_{\boldsymbol{\theta}_{\mathcal{D}_d}}$ is model. The primary objective of dataset distillation task is to generate synthetic data aimed at attaining a specific or minimal performance disparity on the original validation data, when models are trained on the synthetic data and the original dataset, respectively. Thus, we aim to optimize the synthetic data $\mathcal{D}_d$ by:

$$\arg\min_{\mathcal{D}_d, |\mathcal{D}_d|} \left( \sup \left\{ \left| \ell \left( \phi_{\boldsymbol{\theta}_{\mathcal{D}_o}}(\boldsymbol{x}), \boldsymbol{y} \right) - \ell \left( \phi_{\boldsymbol{\theta}_{\mathcal{D}_d}}(\boldsymbol{x}), \boldsymbol{y} \right) \right| \right\}_{(\boldsymbol{x}, \boldsymbol{y}) \sim \mathcal{D}_o} \right) \tag{3}$$

Then, we learn $<\text{data}, \text{label}> \in \mathcal{D}_d$ with the corresponding number of distilled data in each class.

### 2.2 DATASET DISTILLATION ON LARGE-SCALE DATASETS

Currently, the prevailing majority of research studies within dataset distillation mainly employ datasets of a scale up to ImageNet-1K (Cazenavette et al., 2022b; Cui et al., 2023; Yin et al., 2023) as their benchmarking standards. In this section, we show how to construct a strong baseline on ImageNet-21K (this approach is equivalently applicable to ImageNet-1K) by incorporating insights and presented in recent studies, complemented by conventional optimization techniques. Our proposed baseline is demonstrated to achieve state-of-the-art performance. We believe this provides substantial contributions towards understanding the true impact of proposed methodologies and towards assessing the true gap with full original data training. Following prior work in dataset distillation (Yin et al., 2023), we focus on the decoupled training framework to save computation and memory consumption on large-scale ImageNet-21K, and the procedures are listed below:

**Building A Strong Compression Model on ImagNet-21K**. For the squeezing model pretraining, we use a relatively large label smooth of 0.2 together with Cutout (DeVries & Taylor, 2017) and RandAugment (Cubuk et al., 2020), as shown in Appendix B.4. This recipe help achieves ∼2% improvement over the default training Ridnik et al. (2021) on ImageNet-21K, as provided in Table 5.

**Curriculum Training for Better Representation of Synthetic Data**. A well crafted curriculum data augmentation is employed during the synthesis stage to enhance the representational capability of the synthetic data. This step is crucial, serving to enrich the generated images by embedding more knowledge accumulated from the original dataset, thereby making them more informative. Detailed procedures will be further described in the following Section 2.3.

## 2.3 CURRICULUM DATA AUGMENTATION

In SRe²L (Yin et al., 2023) approach, the key of data synthesis revolves around utilizing the gradient information emanating from both the semantic class and the predictions of the pretrained squeezing model, paired with BN distribution matching. Let $(\boldsymbol{x}, \boldsymbol{y})$ be an example $\boldsymbol{x}$ for optimization and its corresponding one-hot label $\boldsymbol{y}$ for the pretrained squeezing model. Throughout synthesis process the squeezing model is frozen for recovering the encoded information ensuring consistency and reliability in the generated data. Let $\mathcal{T}(\boldsymbol{x})$ be the target training distribution from which the data synthesis process should ultimately learn a function of desired trajectory, where $\mathcal{T}$ is a data transformation function to augment input sample to various levels of difficulties. Following Bengio et al. (2009), a weigh $0 \leq W_s(x) \leq 1$ is defined and applied to example $\boldsymbol{x}$ at stage $s$ in the curriculum sequence. The training distribution $D_s(x)$ is:

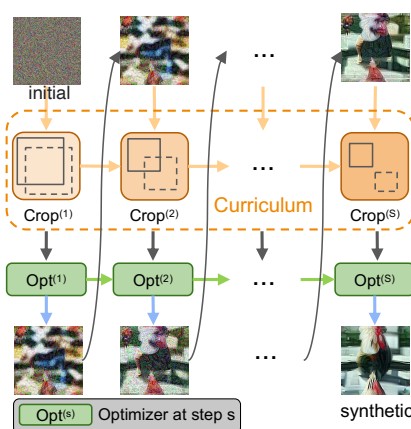

Figure 3: Curriculum data synthesis.

$$D_s(x) \propto W_s(x)\mathcal{T}(x) \quad \forall x \tag{4}$$

In our scenario, since the varying difficulties are governed by the data transformation function $\mathcal{T}$, we can straightforwardly employ $W_s(x) = 1$ across all stages. Consequently, the training distribution solely depends on $\mathcal{T}(x)$ and can be simplified as follows:

$$D(x) \propto \mathcal{T}(x) \quad \forall x \tag{5}$$

By integrating curriculum learning within the data synthesis phase, this procedure can be defined as:

**Definition 1** (Curriculum Data Synthesis)**.** In the data synthesis optimization, the corresponding sequence of distributions $D(x)$ will be a curriculum if there is an increment in the entropy of these distributions, i.e., the difficulty of the transformed input samples escalating and become increasingly challenging for the pre-trained model to predict as the training progresses.

Thus, the key for our curriculum data synthesis becomes how to design $\mathcal{T}(x)$ across different training iterations. The following discusses several strategies to construct this in the curriculum scheme.

**Baseline: Constant Learning (CTL)**. This is the regular training method where all training examples are typically treated equally. Each sample from the training dataset has an equal chance of being transformed in a given batch, assuming no difficulty imbalance or biases across different training iterations.

CTL is straightforward to implement since we do not have to rank or organize examples based on difficulty. In practice, we simply use $\boldsymbol{x}_{\mathcal{T}} \leftarrow \textit{RandomResizedCrop}(\boldsymbol{x}_s, \texttt{min\_crop} = \beta_l, \texttt{max\_crop} = \beta_u)$, where $\beta_l$ and $\beta_u$ are the constant lower and upper bounds of crop scale.

**Curriculum Learning (CL)**. As shown in Algorithm 1, in our CL, data samples are organized based on their difficulty. The difficulty level of the cropped region can be managed by defining the lower and upper scopes for the area ratio of the crop. This enables the assurance that specific crops of the image (small details or broader context) are included in the cropped region. For the difficulty adjustment, the rate at which more difficult examples are introduced and the criteria used to define difficulty are adjusted dynamically as predetermined using the following schedulers.

***Step***. Step scheduler reduces the minimal scale by a factor for every fixed or specified number of iterations.

***Linear***. Linear scheduler starts with a high initial value and decreases it linearly by a factor $\gamma$ to a minimum value over the whole training.

***Cosine***. Cosine scheduler modulates the distribution according to the cosine function of current iteration number, yielding a smoother and more gradual adjustment compared to step-based methods.

As shown in Figure 4, the factor distribution manages the difficulty level of crops with a milestone.

**Data Synthesis by Recovering.** After receiving the transformed input $x_{\mathcal{T}}$, we update it by aligning between the final classification label and intermediate Batch Normalization (BN) statistics from the original data. This stage forces the synthesized images to capture a shape of the original image distribution. The learning goal for this stage can be formulated as follows:

---

**Algorithm 1:** Our **C**urriculum **D**ata **A**ugmentation via *RandomResizedCrop*

---

**Input:** squeezed model $\phi_\theta$, recovery iteration $S$, curriculum milestone $T$, target label $y$, default lower and upper bounds of crop scale $\beta_l$ and $\beta_u$ in *RandomResizedCrop*, decay of lower scale bound $\gamma$

**Output:** synthetic image $x$

**Initialize:** $x_0$ from a standard normal distribution

**for** step $s$ from 0 to $S$-1 **do**

  **if** $s \leq T$ **then**

    $\alpha \leftarrow$

$$\begin{cases} \beta_u & \text{if step} \\ \beta_l + \gamma * (\beta_u - s/T) & \text{if linear} \\ \beta_l + \gamma * (\beta_u + \cos(\pi * s/T))/2 & \text{if cosine} \end{cases}$$

  **else**

    $\alpha \leftarrow \beta_l$

  **end**

  $x_{\mathcal{T}} \leftarrow RandomResizedCrop(x_s, \texttt{min\_crop} = \alpha, \texttt{max\_crop} = \beta_u)$

  $x'_{\mathcal{T}} \leftarrow x_{\mathcal{T}}$ is optimized w.r.t $\phi_\theta$ and $y$ in Eq. 6.

  $x_{s+1} \leftarrow ReverseRandomResizedCrop(x_s, x'_{\mathcal{T}})$

**end**

**return** $x \leftarrow x_S$

---

$$x'_{\mathcal{T}} = \arg\min \ell\left(\phi_\theta\left(x_{\mathcal{T}}\right), y\right) + \mathcal{R}_{\text{reg}} \tag{6}$$

where $\mathcal{R}_{\text{reg}}$ is the regularization term used in Yin et al. (2023), the detailed formulation of it is provided in Appendix D. $\phi_\theta$ is the pretrained squeezing model and will be frozen in this stage. The entire training procedure is illustrated in Figure 3.

## 3 EXPERIMENTS

### 3.1 DATASETS AND IMPLEMENTATION DETAILS

We verify the effectiveness of our approach on various ImageNet scale datasets, including Tiny-ImageNet (Le & Yang, 2015), ImageNet-1K (Deng et al., 2009), and ImageNet-21K (Ridnik et al., 2021). For evaluation, we train models from scratch on synthetic distilled datasets and report the Top-1 accuracy on real validation datasets. Default lower and upper bounds of crop scales $\beta_l$ and $\beta_u$ are 0.08 and 1.0, respectively. The decay $\gamma$ is 0.92. More details are provided in the Appendix B.

### 3.2 CIFAR-100

Comparison results with baseline methods on CIFAR-100 datasets are presented in Table 1. We observe a trend that the validation model's accuracy exhibits a significant im-

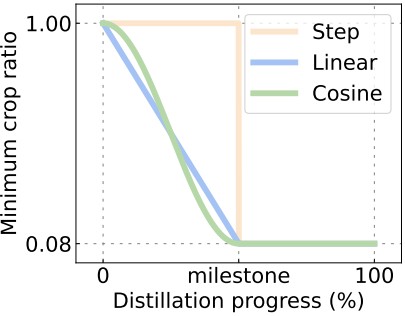

Figure 4: Scheduler of minimum crop ratio in *RandomResizedCrop*.

provement along with the extension of training budgets. Our CDA's best validation accuracy outperforms all baselines under 10 and 50 IPC. And our reported results have the potential to be further improved as training budgets increase. Overall, our CDA method is also applicable to small-scale dataset distillation.

### 3.3 TINY-IMAGENET

Results on the Tiny-ImageNet dataset are detailed in the first group of Table 3. Our CDA surpasses SRE$^2$L with average improvements of 7.7% and 3.4% under IPC 50 and IPC 100 settings across ResNet-{18, 50, 101} validation models, respectively. We provide the result comparison with more

Table 1: Comparison with baseline methods on CIFAR-100.

| CIFAR-100 IPC | DC | DSA | DM | KIP | MTT | CDA (100ep) | CDA (200ep) | CDA (400ep) | CDA (800ep) |
|---|---|---|---|---|---|---|---|---|---|
| 1 | 12.8 | 13.9 | 11.4 | **34.9** | 24.3 | 7.1 | 8.2 | 10.2 | 13.4 |
| 10 | 25.2 | 32.3 | 29.7 | 49.5 | 40.1 | 25.0 | 34.9 | 44.5 | **49.8** |
| 50 | – | 42.8 | 43.6 | – | 47.7 | 48.9 | 56.6 | 60.4 | **64.4** |

baselines in Appendix B.2. Importantly, CDA stands as the inaugural approach to diminish the Top-1 accuracy performance disparity to less than 10% between the distilled dataset employing IPC 100 and the full Tiny-ImageNet, signifying a breakthrough on this dataset.

## 3.4 IMAGENET-1K

Table 2: Constant learning result. $\alpha_l$ and $\alpha_u$ stand for the `min_crop` and `max_crop` parameters in *RandomResizedCrop*. * indicates the reproduced SRe$^2$L result.

| Constant Learning Type \ $\alpha$ | 0.08 | 0.2 | 0.4 | 0.6 | 0.8 | 1.0 |
|---|---|---|---|---|---|---|
| Easy ($\alpha_l = \alpha, \alpha_u = \beta_u$) | 44.90* | 47.88 | 46.34 | 45.35 | 43.48 | 41.30 |
| Hard ($\alpha_l = \beta_l, \alpha_u = \alpha$) | 22.99 | 34.75 | 42.76 | 44.61 | 45.76 | 44.90* |

**Constant Learning (CTL).** We leverage a ResNet-18 and employ 1K recovery synthesis data. As observed in Table 2, the results for exceedingly straightforward or challenging scenarios fall below the reproduced SRe$^2$L baseline accuracy of 44.90%, especially when $\alpha \geq 0.8$ in *easy* and $\alpha \leq 0.4$ in *hard* type. Thus, the results presented in Table 2 suggest that adopting a larger cropped range assists in circumventing extreme scenarios, whether easy or hard, culminating in enhanced performance. A noteworthy observation is the crucial role of appropriate lower and upper bounds for constant learning in boosting validation accuracy. This highlights the importance of employing curriculum data augmentation strategies in data synthesis.

**Reverse Curriculum Learning (RCL).** We use a reverse step scheduler in the RCL experiments, starting with the default cropped range from $\beta_l$ to $\beta_u$ and transitioning at the milestone point to optimize the whole image, shifting from challenging to simpler optimizations. Other settings follow the recovery recipe on ResNet-18 for 1K recovery iterations. Table 4 shows the RCL results, a smaller step milestone indicates an earlier difficulty transition. The findings reveal that CRL does not improve the generated dataset's quality compared to the baseline SRe$^2$L, which has 44.90% accuracy.

Table 4: Ablation of reverse curriculum learning.

| Step Milestone | Accuracy (%) |
|---|---|
| 0.2 | 41.38 |
| 0.4 | 41.59 |
| 0.6 | 42.60 |
| 0.8 | 44.39 |

**Curriculum Learning (CL).** Our CDA experiments follow the recovery recipe of SRe$^2$L's best results for 4K recovery iterations. As illustrated in the second group of Table 3, when compared to the strong baseline SRe$^2$L, CDA enhances the validation accuracy, exhibiting average margins of 6.1%, 4.3%, and 3.2% on ResNet-18, 50, 101 across varying IPC configurations. Furthermore, as shown in Figure 1, the results achieved with our CDA utilizing merely 1K recovery iterations surpass those of SRe$^2$L encompassing the entire 4K iterations. These results substantiate the efficacy and effectiveness of applying CDA in large-scale dataset distillation.

## 3.5 IMAGENET-21K

**Pretraining Results.** Table 5 presents the accuracy for ResNet-18 and ResNet-50 on ImageNet-21K-P, considering varying initial weight configurations. Models pretrained by us and initialized with ImageNet-1K weight exhibit commendable accuracy, showing a 2.0% improvement, while models initialized randomly achieve marginally superior accuracy. We utilize these pretrained models to recover ImageNet-21K data and to assign labels to the synthetic images generated. An intriguing observation is the heightened difficulty in data recovering from pretrained models that are initialized randomly compared to those initialized with ImageNet-1K weight. Thus, our experiments employ CDA specifically on pretrained models that are initialized with ImageNet-1K weight.

**Validation Results.** As illustrated in the final group of Table 3, we perform validation experiments on the distilled ImageNet-21K employing IPC 10 and 20. This yields an extreme compression ratio of $100\times$ and $50\times$. When applying IPC 10, i.e., the models are trained utilizing a distilled dataset that is a mere 1% of the full dataset. Remarkably, validation accuracy surpasses 20% and 30% on ResNet-18 and ResNet-50, 101, respectively. Compared to reproduced SRe$^2$L on ImageNet-21K,

Table 3: Comparison with baseline method SRe$^2$L on various datasets.

| Dataset | IPC | ResNet-18 | | ResNet-50 | | ResNet-101 | |
|---|---|---|---|---|---|---|---|
| | | SRe$^2$L | Ours | SRe$^2$L | Ours | SRe$^2$L | Ours |
| Tiny-ImageNet | 50 | 41.1 | 48.7$^{\uparrow 7.6}$ | 42.2 | 49.7$^{\uparrow 7.5}$ | 42.5 | 50.6$^{\uparrow 8.1}$ |
| | 100 | 49.7 | 53.2$^{\uparrow 3.5}$ | 51.2 | 54.4$^{\uparrow 3.2}$ | 51.5 | 55.0$^{\uparrow 3.5}$ |
| ImageNet-1K | 50 | 46.8 | 53.5$^{\uparrow 6.7}$ | 55.6 | 61.3$^{\uparrow 5.7}$ | 57.6 | 61.6$^{\uparrow 4.0}$ |
| | 100 | 52.8 | 58.0$^{\uparrow 5.2}$ | 61.0 | 65.1$^{\uparrow 4.1}$ | 62.8 | 65.9$^{\uparrow 3.1}$ |
| | 200 | 57.0 | 63.3$^{\uparrow 6.3}$ | 64.6 | 67.6$^{\uparrow 3.0}$ | 65.9 | 68.4$^{\uparrow 2.5}$ |
| ImageNet-21K | 10 | 18.5 | 22.6$^{\uparrow 4.1}$ | 27.4 | 32.4$^{\uparrow 5.0}$ | 27.3 | 34.2$^{\uparrow 6.9}$ |
| | 20 | 20.5 | 26.4$^{\uparrow 5.9}$ | 29.5 | 35.3$^{\uparrow 5.8}$ | 31.8 | 36.1$^{\uparrow 4.3}$ |

Table 5: Accuracy of ResNet-{18, 50} on ImageNet-21K-P.

| Model | Initial Weight | Top-1 Acc. (%) | Top-5 Acc. (%) |
|---|---|---|---|
| ResNet-18 (Ours) | ImageNet-1K | 38.1 | 67.2 |
| | Random | 38.5 | 67.8 |
| Ridnik et al. (2021) | ImageNet-1K | 42.2 | 72.0 |
| ResNet-50 (Ours) | ImageNet-1K | 44.2$^{\uparrow 2.0}$ | 74.6$^{\uparrow 2.6}$ |
| | Random | 44.5$^{\uparrow 2.3}$ | 75.1$^{\uparrow 3.1}$ |

our approach attains an elevation of 5.3% on average under IPC 10/20. This achievement not only highlights the efficacy of our approach in maintaining dataset essence despite high compression but also showcases the potential advancements in accuracy over existing methods.

## 3.6 ABLATION

**Curriculum Scheduler.** To schedule the curriculum learning, we present three distinct types of curriculum schedulers, *step*, *linear*, and *cosine* to manipulate the lower bounds on data cropped augmentation. As illustrated in Figure 4, the dataset distillation progress is divided into two phases by a milestone. It is observed that both *linear* and *cosine* with continuous decay manifest robustness across diverse milestone configurations and reveal a trend of enhancing accuracy performance when the milestone is met at a later phase, as shown in Table 5. Moreover, *cosine* marginally outperforms *linear* in terms of accuracy towards the end. Consequently, we choose to implement the *cosine* scheduler, assigning a milestone percentage of 1.0, to modulate the minimum crop ratio adhering to the principles of curriculum learning throughout the progression of synthesis.

**Batch Size in Post-training.** We perform an ablation study to assess the influence of utilizing smaller batch sizes on the generalization performance of models when the synthetic data is limited.

Table 6: Ablation on batch size in validation recipes.

| Batch Size | Accuracy (%) |
|---|---|
| 128 | 20.79 |
| 64 | 21.85 |
| 32 | 22.54 |
| 16 | **22.75** |
| 8 | 22.41 |

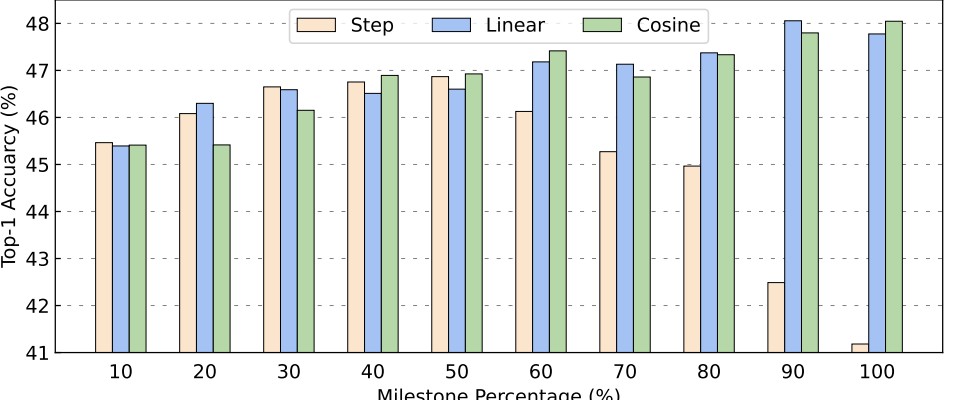

Figure 5: Ablation study on three different schedulers with varied milestone settings.

Table 7: ImageNet-1K Top-1 on cross-model generation. Distilled datasets consist of 50 IPC.

| Recovery Model | Validation Model | | | | | | |
|---|---|---|---|---|---|---|---|
| | R18 | R50 | R101 | DenseNet-121 | RegNet-Y-8GF | ConvNeXt-Tiny | DeiT-Tiny |
| ResNet-18 (SRe$^2$L) | 46.80 | 55.60 | 57.60 | 49.74 | 60.34 | 53.53 | 15.41 |
| ResNet-18 (Ours) | 53.45 | 61.26 | 61.57 | 57.35 | 63.22 | 62.58 | 31.95 |
| DenseNet-121 (Ours) | 49.52 | 58.22 | 56.53 | 53.72 | 61.99 | 60.83 | 22.87 |

Table 8: ImageNet-21K Top-1 on cross-model generation. Distilled datasets consist of 20 IPC.

| Recovery Model | Validation Model | | | | |
|---|---|---|---|---|---|
| | ResNet-18 | ResNet-50 | ResNet-101 | DenseNet-121 | RegNet-Y-8GF |
| ResNet-18 | 26.42 | 35.32 | 36.12 | 28.66 | 36.13 |
| ResNet-50 | 22.95 | 34.14 | 35.46 | 26.01 | 34.93 |

We report results on the distilled ImageNet-21K from ResNet-18. In Table 6, a rise in validation accuracy is observed as batch size reduces, peaking at 16. This suggests that smaller batch sizes enhance performance on small-scale synthetic datasets. However, this leads to more frequent data loading and lower GPU utilization in our case, extending training times. To balance training time with performance, we chose a batch size of 32 for our experiments.

## 3.7 ANALYSIS

**Cross-Model Generalization**. The challenge of ensuring distilled datasets generalize effectively across models unseen during the recovery phase remains significant, as in prior approaches (Zhao et al., 2020; Cazenavette et al., 2022a), synthetic images were optimized to overfit the recovery model. In Table 7, we deploy our ImageNet-1K distilled datasets to train validation models, and we attain over 60% Top-1 accuracy with most of these models. Additionally, our performance in Top-1 accuracy surpasses that of SRe$^2$L across all validation models spanning various architectures. It is remarkable that the distilled datasets exhibit reduced dependency on specific recovery models, thereby alleviating the issues associated with overfitting optimization. Table 8 supports further empirical substantiation of the CDA's efficacy in the distillation of large-scale ImageNet-21K datasets. More validation models are included in Table 20 of Appendix.

**Impact of Curriculum**. To study the curriculum's advantage on synthetic image characteristics, we evaluate the Top-1 accuracy on CDA, SRe$^2$L and real ImageNet-1K training set, using mean of random 10-crop and global images. We employ PyTorch's pre-trained MobileNet-V2 for classifying these images. As shown in Table 9, CDA images closely resemble real ImageNet images in prediction accuracies, better than SRe$^2$L. Consequently, using curriculum data augmentation improves global image prediction and reduces bias and overfitting in post-training on simpler, cropped images of SRe$^2$L.

Table 9: Classification accuracy of synthetic and real images using MobileNet-V2.

| Top-1 (%) | Dataset | | |
|---|---|---|---|
| | SRe$^2$L | CDA (ours) | Real |
| global | 79.34 | 81.25 | 82.16 |
| cropped | 87.48 | 82.44 | 72.73 |

**Visualization and Discussion**. Figure 6 provides a comparative visualization of the gradient synthetic images at recovery steps of $\{100, 500, 1,000, 2,000\}$ to illustrate the differences between SRe$^2$L and our CDA within the dataset distillation process. SRe$^2$L images in the upper line exhibit a significant amount of noise, indicating a slow recovery progression in the early recovery stage. On the contrary, due to the mostly entire image optimization in the early stage, CDA images in the lower

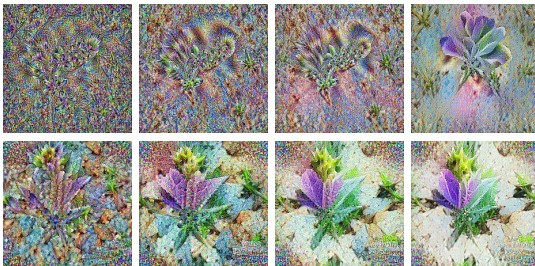

Figure 6: Synthetic ImageNet-21K images (*Plant*).

line can establish the layout of the entire image and reduce noise rapidly. And the final synthetic images contain more visual information directly related to the target class *Plant*. Therefore, the comparison highlights CDA's ability to synthesize images with enhanced visual coherence to the target class, offering a more efficient recovery process. More visualizations are provided in our appendix.

**Training cost comparison with SRe$^2$L**. We highlight that there is no additional training cost incurred between our CDA and SRe$^2$L (Yin et al., 2023) when the recovery iterations are the same.

Specifically, it takes about 55 hours on $4\times$ RTX 4090 GPUs to generate our distilled ImageNet-21K with 20 IPC and the peak GPU memory utilization is 15GB. More details are in Appendix C.

### 3.8 APPLICATION: CONTINUAL LEARNING

Distilled datasets, comprising high-semantic images, possess a boosted representation capacity compared to the original datasets. This attribute can be strategically harnessed to combat catastrophic forgetting in continual learning. We have further validated the effectiveness of our introduced CDA within various continual learning scenarios. Following the setting outlined in SRe$^2$L (Yin et al., 2023), we conducted 5-step and 10-step class-incremental experiments on Tiny-ImageNet, aligning our results against the baseline SRe$^2$L and a randomly selected subset on Tiny-ImageNet for comparative analysis. Illustrated in Figure 7, our CDA distilled dataset notably surpasses SRe$^2$L, exhibiting an average advantage of 3.8% and 4.5% on 5-step and 10-step class-incremental learning assignments respectively. This demonstrates the substantial benefits inherent in the deployment of CDA, particularly in mitigating the complexities associated with continual learning.

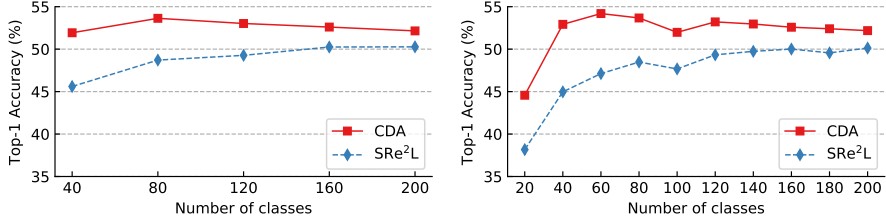

Figure 7: 5-step and 10-step class-incremental learning on Tiny-ImageNet.

## 4 RELATED WORK

Dataset distillation strives to form a compact, synthetic dataset, retaining crucial information from the original large-scale dataset. This approach facilitates easier handling, reduces training time, and aims for performance comparable to using the full dataset. Prior solutions typically fall under four categories: *Meta-Model Matching* optimizes for model transferability on distilled data, with an outer-loop for synthetic data updates, and an inner-loop for network training, such as DD (Wang et al., 2020), KIP (Nguyen et al., 2021), RFAD (Loo et al., 2022), FRePo (Zhou et al., 2022) and LinBa (Deng & Russakovsky, 2022); *Gradient Matching* performs a one-step distance matching between models, such as DC (Zhao et al., 2020), DSA (Zhao & Bilen, 2021), DCC (Lee et al., 2022) and IDC (Kim et al., 2022b); *Distribution Matching* directly matches the distribution of original and synthetic data with a single-level optimization, such as DM (Zhao & Bilen, 2023), CAFE (Wang et al., 2022), HaBa (Liu et al., 2022a), KFS (Lee et al., 2022); *Trajectory Matching* matches the weight trajectories of models trained on original and synthetic data in multiple steps, methods include MTT (Cazenavette et al., 2022b) and TESLA (Cui et al., 2023). Bengio et al. (2009) introduced the idea of *Curriculum Learning*, proposing that training models with a curriculum can help in better optimization and can lead to better generalization. Subsequent works (Jiang et al., 2018; Ren et al., 2018; Hacohen & Weinshall, 2019) have explored various strategies for defining and designing curricula, such as self-paced learning (Kumar et al., 2010), where the model itself determines the learning pace and difficulty progression.

## 5 CONCLUSION

We have presented a new CDA framework focused on curriculum-based data synthesis for large-scale dataset distillation. Our approach involves a practical framework with detailed pertaining for compressing knowledge, data synthesis for recovery, and post-training recipes. The proposed approach enables the distillation of ImageNet-21K to $50\times$ smaller while maintaining competitive accuracy levels. In regular benchmarks, such as ImageNet-1K, our approach has demonstrated superior performance, surpassing all prior state-of-the-art methods by substantial margins. We further show the capability of our synthetic data on cross-model generalization and continual learning. Given the escalating scale of both models and datasets in recent years, the imperativeness of dataset distillation for large-scale datasets and large-scale models has gained unprecedented prominence and urgency. We hope our contributions in this work will infuse novel insights and pave new avenues within this domain, encouraging further exploration and development in the field of large-scale dataset distillation. Our future work will focus on distilling more modalities like language and speech.

**Reproducibility Statement**. We provide our detailed training recipes in Appendix B. Our synthetic data is available anonymously at link and our code is also provided in submission.

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

APPENDIX

In the appendix, we provide additional details that were not included in the main text:

- Section A: Dataset details.
- Section B: Implementation details.
- Section C: Computational Cost.
- Section D: Formulation of Regularization and More Discussion.
- Section E: More visualization of synthetic data.

## A    DATASETS DETAILS

We conduct experiments on three ImageNet scale datasets, Tiny-ImageNet (Le & Yang, 2015), ImageNet-1K (Deng et al., 2009), and ImageNet-21K (Ridnik et al., 2021). The dataset details are as follows:

- CIFAR-100 dataset composes 500 training images per class, each with a resolution of 32×32 pixels, across 100 classes.
- Tiny-ImageNet dataset is derived from ImageNet-1K and consists of 200 classes. Within each category, there are 500 images with a uniform 64×64 resolution.
- ImageNet-1K dataset comprises 1,000 classes and 1,281,167 images in total. We resize all images into standard 224×224 resolution during the data loading stage.
- The original ImageNet-21K dataset is an extensive visual recognition dataset containing 21,841 classes and 14,197,122 images. We use ImageNet-21K-P (Ridnik et al., 2021) which utilizes data processing to remove infrequent classes and resize all images to 224×224 resolution. After data processing, ImageNet-21K-P dataset consists of 10,450 classes and 11,060,223 images.

## B    IMPLEMENTATION DETAILS

### B.1    CIFAR-100

**Hyper-parameter Setting.**

We train a modified ResNet-18 model (He et al., 2020) on CIFAR-100 training data with a Top-1 accuracy of 78.16% using the parameter setting in Table 10a. The well-trained model serves as the recovery model under the recovery setting in Table 10b.

Table 10: Parameter setting on CIFAR-100.

| (a) Squeezing/validation setting. | | (b) Recovery setting. | |
|---|---|---|---|
| config | value | config | value |
| optimizer | SGD | $\alpha_{BN}$ | 0.01 |
| base learning rate | 0.1 | optimizer | Adam |
| momentum | 0.9 | base learning rate | 0.25 |
| weight decay | 5e-4 | optimizer momentum | $\beta_1, \beta_2 = 0.5, 0.9$ |
| batch size | 128 (squeeze) / 8 (val) | batch size | 100 |
| learning rate schedule | cosine decay | learning rate schedule | cosine decay |
| training epoch | 200 (squeeze) / 100 (val) | recovery iteration | 1,000 |
| augmentation | RandomResizedCrop | augmentation | RandomResizedCrop |

Due to the low resolution of CIFAR images, the default lower bound $\beta_l$ needs to be raised from 0.08 (ImageNet setting) to a higher reasonable value in order to avoid the training inefficiency caused by extremely small cropped areas with little information. Thus, we conducted the ablation to select the optimal value for the default lower bound $\beta_l$ in RandomResizedCrop operations in Table 11. We choose 0.4 as the default lower bound $\beta_l$ in Algorithm 1 to exhibit the best distillation performance on CIFAR-100. We adopt a small batch size value of 8 and extend the training budgets in the following validation stage, which aligns with the strong training recipe on inadequate datasets.

Table 11: Ablation on the lower bound $\beta_l$ setting in distilling CIFAR-100

| default lower bound $\beta_l$ | 0.08 | 0.2 | 0.4 | 0.6 | 0.8 | 1.0 |
|---|---|---|---|---|---|---|
| validation accuracy (800ep) (%) | 58.5 | 62.14 | **64.4** | 63.36 | 61.65 | 54.43 |

## B.2 TINY-IMAGENET

**Hyper-parameter Setting.** We train a modified ResNet-18 model (He et al., 2020) on Tiny-ImageNet training data with the parameter setting in Table 12a and use the well-trained ResNet-18 model with a Top-1 accuracy of 59.47% as a recovery model for CDA. The recovery setting is provided in Table 12b.

Table 12: Parameter setting on Tiny-ImageNet.

(a) Squeezing/validation setting.

| config | value |
|---|---|
| optimizer | SGD |
| base learning rate | 0.2 |
| momentum | 0.9 |
| weight decay | 1e-4 |
| batch size | 256 (squeeze) / 64 (val) |
| learning rate schedule | cosine decay |
| training epoch | 50 (squeeze) / 100 (val) |
| augmentation | RandomResizedCrop |

(b) Recovery setting.

| config | value |
|---|---|
| $\alpha_{BN}$ | 1.0 |
| optimizer | Adam |
| base learning rate | 0.1 |
| optimizer momentum | $\beta_1, \beta_2 = 0.5, 0.9$ |
| batch size | 100 |
| learning rate schedule | cosine decay |
| recovery iteration | 4,000 |
| augmentation | RandomResizedCrop |

**Small IPC Setting Comparison.** Table 13 presents the result comparison between our CDA with DM (Zhao & Bilen, 2023) and MTT (Cazenavette et al., 2022b). Consider that our approach is a decoupled process of dataset compression followed by recovery through gradient updating. It is well-suited to large-scale datasets but less so for small IPC values. As anticipated, there is no advantage when IPC value is extremely low, such as IPC = 1. However, when the IPC is increased slightly, our method demonstrates considerable benefits on accuracy over other counterparts. Furthermore, we emphasize that our approach yields substantial improvements when afforded a larger training budget, i.e., more training epochs.

Table 13: Comparison with baseline methods on Tiny-ImageNet.

| Tiny-ImageNet IPC | DM | MTT | CDA (200ep) | CDA (400ep) | CDA (800ep) |
|---|---|---|---|---|---|
| 1 | 3.9 | **8.8** | $2.38 \pm 0.08$ | $2.82 \pm 0.06$ | $3.29 \pm 0.26$ |
| 10 | 12.9 | 23.2 | $30.41 \pm 1.53$ | $37.41 \pm 0.02$ | $\mathbf{43.04 \pm 0.26}$ |
| 20 | – | – | $43.93 \pm 0.20$ | $47.76 \pm 0.19$ | $\mathbf{50.46 \pm 0.14}$ |
| 50 | 24.1 | 28.0 | $50.26 \pm 0.09$ | $51.52 \pm 0.17$ | $\mathbf{55.50 \pm 0.18}$ |

**Continual Learning.** We adhere to the continual learning codebase outlined in Zhao et al. (2020) and validate provided $SRe^2L$ and our CDA distilled Tiny-ImageNet dataset under IPC 100 as illustrated in Figure 7. Detailed values are presented in the Table 14 and Table 15.

Table 14: 5-step class-incremental learning on Tiny-ImageNet. This table complements details in the left subfigure of Figure 7.

| # class | 40 | 80 | 120 | 160 | 200 |
|---|---|---|---|---|---|
| $SRe^2L$ | 45.60 | 48.71 | 49.27 | 50.25 | 50.27 |
| CDA (ours) | 51.93 | 53.63 | 53.02 | 52.60 | 52.15 |

Table 15: 10-step class-incremental learning on Tiny-ImageNet. This table complements details in the right subfigure of Figure 7.

| # class | 20 | 40 | 60 | 80 | 100 | 120 | 140 | 160 | 180 | 200 |
|---|---|---|---|---|---|---|---|---|---|---|
| $SRe^2L$ | 38.17 | 44.97 | 47.12 | 48.48 | 47.67 | 49.33 | 49.74 | 50.01 | 49.56 | 50.13 |
| CDA (ours) | 44.57 | 52.92 | 54.19 | 53.67 | 51.98 | 53.21 | 52.96 | 52.58 | 52.40 | 52.18 |

### B.3 IMAGENET-1K

**Hyper-parameter Setting.** We employ PyTorch off-the-shelf ResNet-18 and DenseNet-121 with the Top-1 accuracy of {69.79%, 74.43%} which are trained with the official recipe in Table 16a. And the recovery settings are provided in Table 16c, and it is noteworthy that we tune and set distinct parameters $\alpha_{BN}$ and learning rate for different recovery models in Table 16d. Then, we employ ResNet-{18, 50, 101, 152} (He et al., 2016), DenseNet-121 (Huang et al., 2017), RegNet (Radosavovic et al., 2020), ConvNeXt (Liu et al., 2022b), and DeiT-Tiny (Touvron et al., 2021) as validation models to evaluate the cross-model generalization on distilled ImageNet-1K dataset under the validation setting in Table 16b.

Table 16: Parameter setting on ImageNet-1K.

(a) Squeezing setting.

| config | value |
|---|---|
| optimizer | SGD |
| base learning rate | 0.1 |
| momentum | 0.9 |
| weight decay | 1e-4 |
| batch size | 256 |
| lr step size | 30 |
| lr gamma | 0.1 |
| training epoch | 90 |
| augmentation | RandomResizedCrop |

(b) Validation setting.

| config | value |
|---|---|
| optimizer | AdamW |
| base learning rate | 1e-3 |
| weight decay | 1e-2 |
| batch size | 128 |
| learning rate schedule | cosine decay |
| training epoch | 300 |
| augmentation | RandomResizedCrop |

(c) Shared recovery setting.

| config | value |
|---|---|
| optimizer | Adam |
| optimizer momentum | $\beta_1, \beta_2 = 0.5, 0.9$ |
| batch size | 100 |
| learning rate schedule | cosine decay |
| augmentation | RandomResizedCrop |

(d) Model-specific recovery setting.

| config | ResNet-18 | DenseNet-121 |
|---|---|---|
| $\alpha_{BN}$ | 0.01 | 0.01 |
| base learning rate | 0.25 | 0.5 |
| recovery iteration | 1,000 / 4,000 | 1,000 |

**Histogram Values.** The histogram data of ImageNet-1K comparison with SRe2L in Figure 1 can be conveniently found in the following Table 17 for reference.

Table 17: ImageNet-1K comparison with SRe$^2$L. This table complements details in Figure 1.

| Method \ Validation Model | ResNet-18 | ResNet-50 | ResNet-101 | DenseNet-121 | RegNet-Y-8GF |
|---|---|---|---|---|---|
| SRe$^2$L (4K) | 46.80 | 55.60 | 57.59 | 49.74 | 60.34 |
| Our CDA (1K) | 52.88 | 60.70 | 61.10 | 57.26 | 62.94 |
| Our CDA (4K) | 53.45 | 61.26 | 61.57 | 57.35 | 63.22 |

To conduct the ablation studies efficiently in Table 2, Table 4 and Figure 5, we recover the data for 1,000 iterations and validate the distilled dataset with a batch size of 1,024, keeping other settings the same as Table 16. Detailed values of the ablation study on schedulers are provided in Table 18.

Table 18: Ablation study on three different schedulers with varied milestone settings. This table complements details in Figure 5.

| Scheduler \ Milestone | 0.1 | 0.2 | 0.3 | 0.4 | 0.5 | 0.6 | 0.7 | 0.8 | 0.9 | 1 |
|---|---|---|---|---|---|---|---|---|---|---|
| Step | 45.46 | 46.08 | 46.65 | 46.75 | 46.87 | 46.13 | 45.27 | 44.97 | 42.49 | 41.18 |
| Linear | 45.39 | 46.30 | 46.59 | 46.51 | 46.60 | 47.18 | 47.13 | 47.37 | 48.06 | 47.78 |
| Cosine | 45.41 | 45.42 | 46.15 | 46.90 | 46.93 | 47.42 | 46.86 | 47.33 | 47.80 | 48.05 |

**Ablation study on relabeling.** We employ ResNet-18 and DenseNet-121 to relabel the distilled datasets recovered from DenseNet-121 and the validation results are present in Table 19. The results demonstrate that employing DenseNet-121 as the data labeling yields superior results for Densenet-121 and RegNet-Y-8GF validation models. Conversely, for other validation models, employing

ResNet-18 as the data labeling is more effective. In the last row of ImageNet-21K cross-model generation results in Table 8, we report the superior results between these two relabeling models, highlighted in Table 19.

Table 19: Ablation study on relabeling model for distilled datasets recovered from DenseNet-121.

| Relabeling Model | Validation Model | | | | | | |
|---|---|---|---|---|---|---|---|
| | R18 | R50 | R101 | DenseNet-121 | RegNet-Y-8GF | ConvNeXt-Tiny | DeiT-Tiny |
| ResNet-18 | 49.52 | 58.22 | 56.53 | 53.27 | 60.93 | 60.83 | 22.87 |
| DenseNet-121 | 42.23 | 56.48 | 54.58 | 53.72 | 61.99 | 59.73 | 13.14 |

**Cross-Model Generalization.** To supplement the validation models in Table 7, including more different architecture models to evaluate the cross-architecture performance. We have conducted validation experiments on a broad range of models, including SqueezeNet, MobileNet, EfficientNet, MNASNet, ShuffleNet, ResMLP, AlexNet, DeiT-Base, and VGG family models. These validation models are selected from a wide variety of architectures, encompassing a vast range of parameters, shown in Table 20. In the upper group of the table, the selected models are relatively small and efficient. There is a trend that its validation performance improves as the number of model parameters increases. In the lower group, we validated earlier models AlexNet and VGG. These models also show a trend of performance improvement with increasing size, but due to the simplicity of early model architectures, such as the absence of residual connections, their performance is inferior compared to more recent models. Additionally, we evaluated our distilled dataset on ResMLP, which is based on MLPs, and the DeiT-Base model, which is based on transformers. In summary, the distilled dataset created using our CDA method demonstrates strong validation performance across a wide range of models, considering both architectural diversity and parameter size.

Table 20: ImageNet-1K Top-1 on cross-model generation. Our CDA dataset consists of 50 IPC.

| Model | SqueezeNet | MobileNet | EfficientNet | MNASNet | ShuffleNet | ResMLP |
|---|---|---|---|---|---|---|
| #Params (M) | 1.2 | 3.5 | 5.3 | 6.3 | 7.4 | 30.0 |
| accuracy (%) | 19.70 | 49.76 | 55.10 | 55.66 | 54.69 | 54.18 |

| Model | AlexNet | DeiT-Base | VGG-11 | VGG-13 | VGG-16 | VGG-19 |
|---|---|---|---|---|---|---|
| #Params (M) | 61.1 | 86.6 | 132.9 | 133.0 | 138.4 | 143.7 |
| accuracy (%) | 14.60 | 30.27 | 36.99 | 38.60 | 42.28 | 43.30 |

### B.4 IMAGENET-21K

**Hyper-parameter Setting.** ImageNet-21K-P (Ridnik et al., 2021) proposes two training recipes to train ResNet-{18, 50} models. One way is to initialize the models from well-trained ImageNet-1K weight and train on ImageNet-21K-P for 80 epochs, another is to train models with random initialization for 140 epochs, as shown in Table 21a. The accuracy metrics on both training recipes are reported in Table 5. In our experiments, we utilize the pretrained ResNet-{18, 50} models initialized by ImageNet-1K weight with the Top-1 accuracy of {38.1%, 44.2%} as recovery model. And the recovery setting is provided in Table 21c. Then, we evaluate the quality of the distilled ImageNet-21K dataset on ResNet-{18, 50, 101} validation models under the validation setting in Table 21b. To accelerate the ablation study on the batch size setting in Table 6, we train the validation model ResNet-18 for 140 epochs.

## C COMPUTATIONAL COST

For ImageNet-1K, we use the off-the-shelf PyTorch's pretrained models as the squeezing model freely. In the recovery phase, to generate the distilled ImageNet-1K with IPC of 50, it takes about 29 hours on a single A100 (40G) GPU and the peak GPU memory utilization is 6.7GB.

For ImageNet-21K, in the squeezing phase, we follow the official scripts in the ImageNet-21K-P dataset (Winter 21 version). It takes 32 hours to squeeze the original training dataset into a ResNet-18 model on $4\times$ A100 (40G) GPUs. In the recovery phase, we generate ImageNet-21K images with

Table 21: Parameter setting on ImageNet-21K.

(a) Squeezing setting.

| config | value |
|---|---|
| optimizer | Adam |
| base learning rate | 3e-4 |
| weight decay | 1e-4 |
| batch size | 1,024 |
| learning rate schedule | cosine decay |
| label smooth | 0.2 |
| training epoch | 80/140 |
| augmentation | CutoutPIL, RandAugment |

(b) Validation setting.

| config | value |
|---|---|
| optimizer | AdamW |
| base learning rate | 2e-3 |
| weight decay | 1e-2 |
| batch size | 32 |
| learning rate schedule | cosine decay |
| label smooth | 0.2 |
| training epoch | 300 |
| augmentation | CutoutPIL, RandomResizedCrop |

(c) Recovery setting.

| config | value |
|---|---|
| $\alpha_{\text{BN}}$ | 0.25 |
| optimizer | Adam |
| base learning rate | 0.05 (ResNet-18), 0.1 (ResNet-50) |
| optimizer momentum | $\beta_1, \beta_2 = 0.5, 0.9$ |
| batch size | 100 |
| learning rate schedule | cosine decay |
| recovery iteration | 2,000 |
| augmentation | RandomResizedCrop |

1 IPC on a single RTX 4090 GPU, taking 11 hours on average. To generate the distilled ImageNet-21K with IPC of 20, it takes about 55 hours on $4\times$ RTX 4090 GPUs, and the peak GPU memory utilization is 15GB.

## D  FORMULATION OF REGULARIZATION AND MORE DISCUSSION

The formulation of $\mathcal{R}_{\text{reg}}$ in the main paper is:

$$
\begin{aligned}
\mathcal{R}_{\text{reg}}\left(\boldsymbol{x}'\right) &= \sum_k \left\|\mu_k\left(\boldsymbol{x}'\right) - \mathbb{E}\left(\mu_k \mid \mathcal{D}_o\right)\right\|_2 + \sum_k \left\|\sigma_l^2\left(\boldsymbol{x}'\right) - \mathbb{E}\left(\sigma_k^2 \mid \mathcal{D}_o\right)\right\|_2 \\
&\approx \sum_k \left\|\mu_k\left(\boldsymbol{x}'\right) - \mathbf{BN}_k^{\text{RM}}\right\|_2 + \sum_k \left\|\sigma_k^2\left(\boldsymbol{x}'\right) - \mathbf{BN}_k^{\text{RV}}\right\|_2
\end{aligned}
\tag{7}
$$

where $k$ is the index of BN layer, $\mu_l\left(\boldsymbol{x}'\right)$ and $\sigma_l^2\left(\boldsymbol{x}'\right)$ are the channel-wise mean and variance in current batch data. $\mathbf{BN}_k^{\text{RM}}$ and $\mathbf{BN}_k^{\text{RV}}$ are mean and variance in the pre-trained model at $k$-th BN layer, which are globally counted.

**Advantages of Curriculum Data Synthesis**. The proposed CDA enjoys several advantages: (1) Stabilized training: Curriculum synthesis can provide a more stable training process as it reduces drastic loss fluctuations that can occur when the learning procedure encounters a challenging sample early on. (2) Better generalization: By gradually increasing the difficulty, the synthetic data can potentially achieve better generalization on diverse model architectures in post-training. It reduces the chance of the synthesis getting stuck in poor local minima early in the training process. (3) Avoid overfitting: By ensuring that the synthetic data is well-tuned on simpler examples before encountering outliers or more challenging data, there is a potential to reduce overfitting. This is because the foundational concepts are solidified before the synthetic data tries to over-adjust for rare or complicated samples. This is examined empirically in our experiments.

**Post-training on Larger Models with Stronger Training Recipes**. Prior studies, such as TESLA (Cui et al., 2023), has encountered difficulties, particularly, a decline in accuracy when utilizing models of larger scale. This suggests that the synthetic data used is potentially inadequate for training larger models. Conversely, the data we generated show improvement with the use of larger models combined with enhanced post-training methodologies, displaying promise when applied to larger datasets in distillation processes. We have also observed that maintaining a smaller batch size is crucial for post-training on synthetic data to achieve commendable accuracy. This is

attributed to the *Generalization Gap* (Keskar et al., 2016; Hoffer et al., 2017), which suggests that when there is a deficiency in the total training samples, the model's capacity to generalize to new, unseen data is not robust. Employing smaller batch sizes while training on synthetic data allows models to explore the loss landscape more meticulously before converging to an optimal minimum.

## E  VISULIZATION

We provide additional comparisons of four groups of visualizations on synthetic ImageNet-21K images at recovery steps of $\{100, 500, 1{,}000, 1{,}500, 2{,}000\}$ between SRe$^2$L (upper) and CDA (lower) in Figure 8. The chosen target classes are *Benthos*, *Squash Rackets*, *Marine Animal*, and *Scavenger*.

In addition, we present our CDA's synthetic ImageNet-1K images in Figure 9 and ImageNet-21K images in Figure 10 and Figure 11.

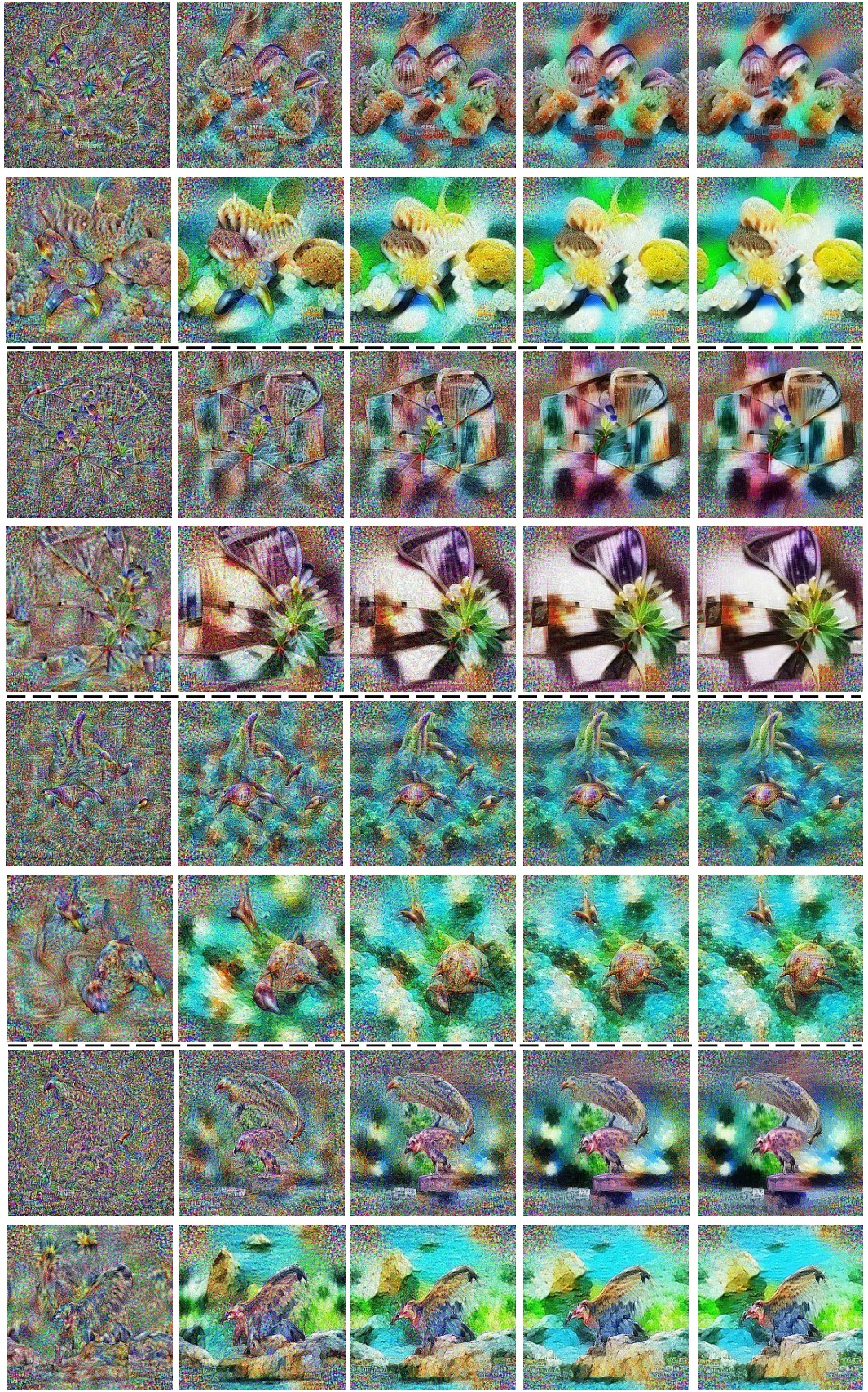

Figure 8: Synthetic ImageNet-21 data visualization comparison.

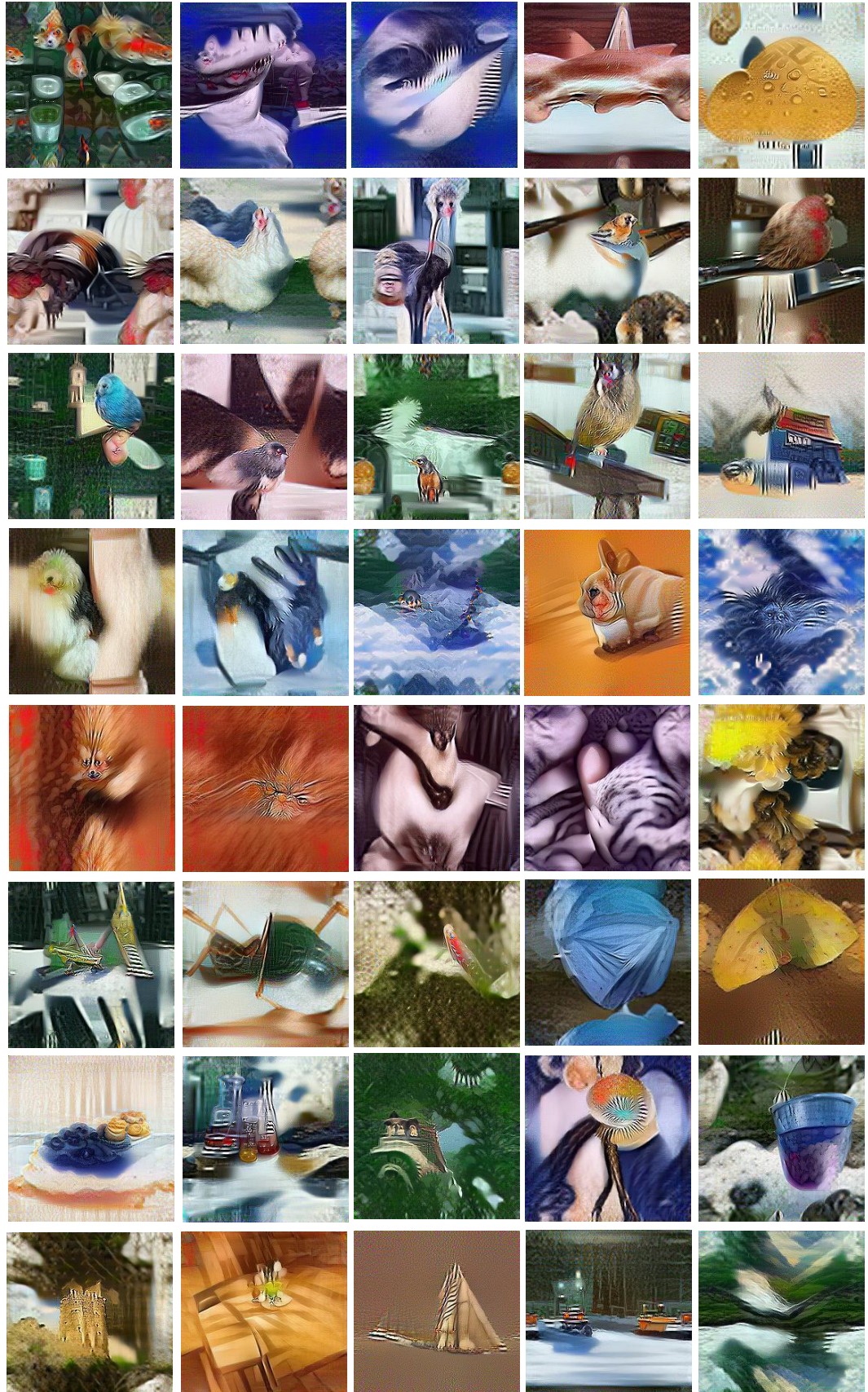

Figure 9: Synthetic ImageNet-1K data visualization from CDA.

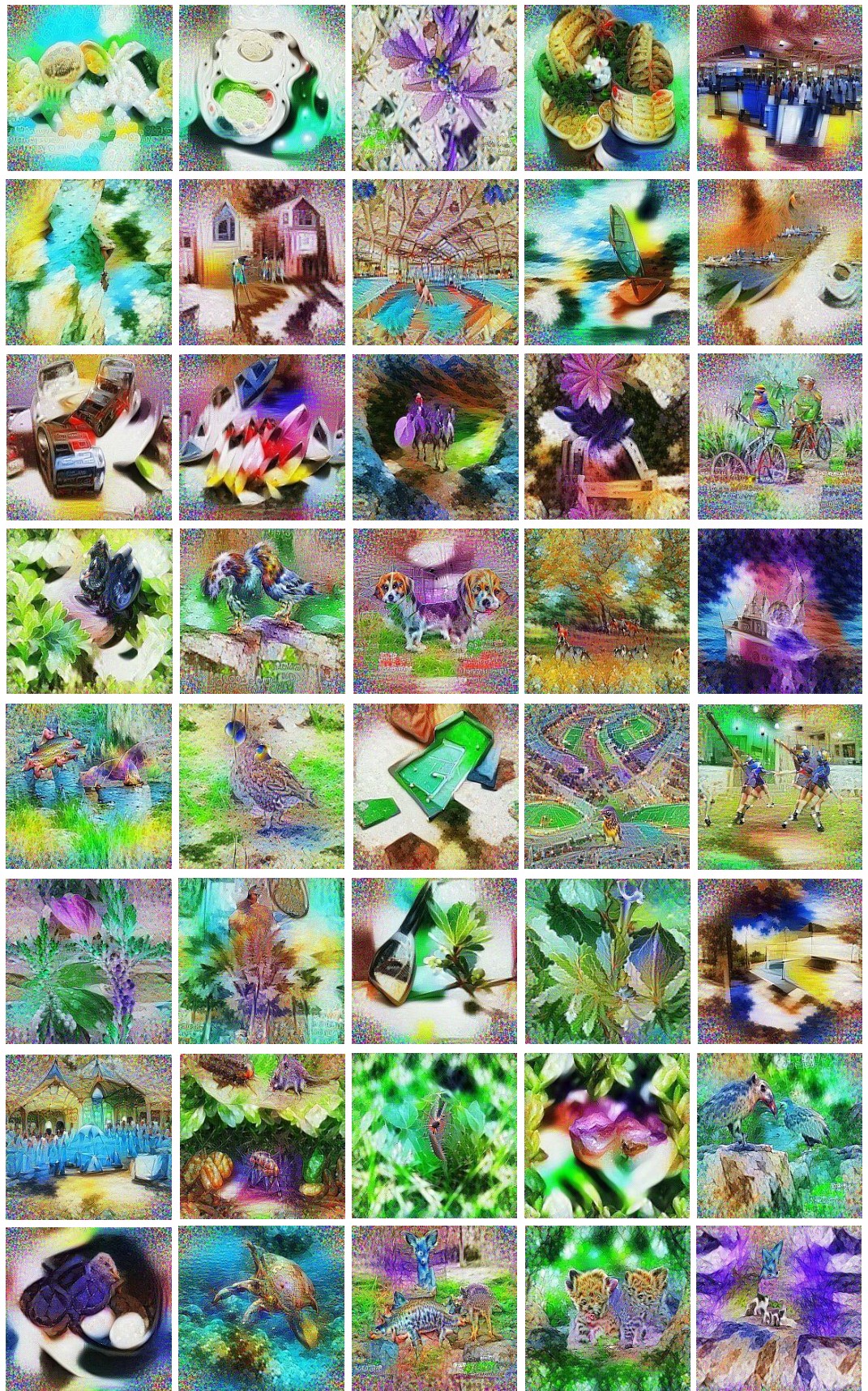

Figure 10: Synthetic ImageNet-21K data distilled from ResNet-18 by CDA.

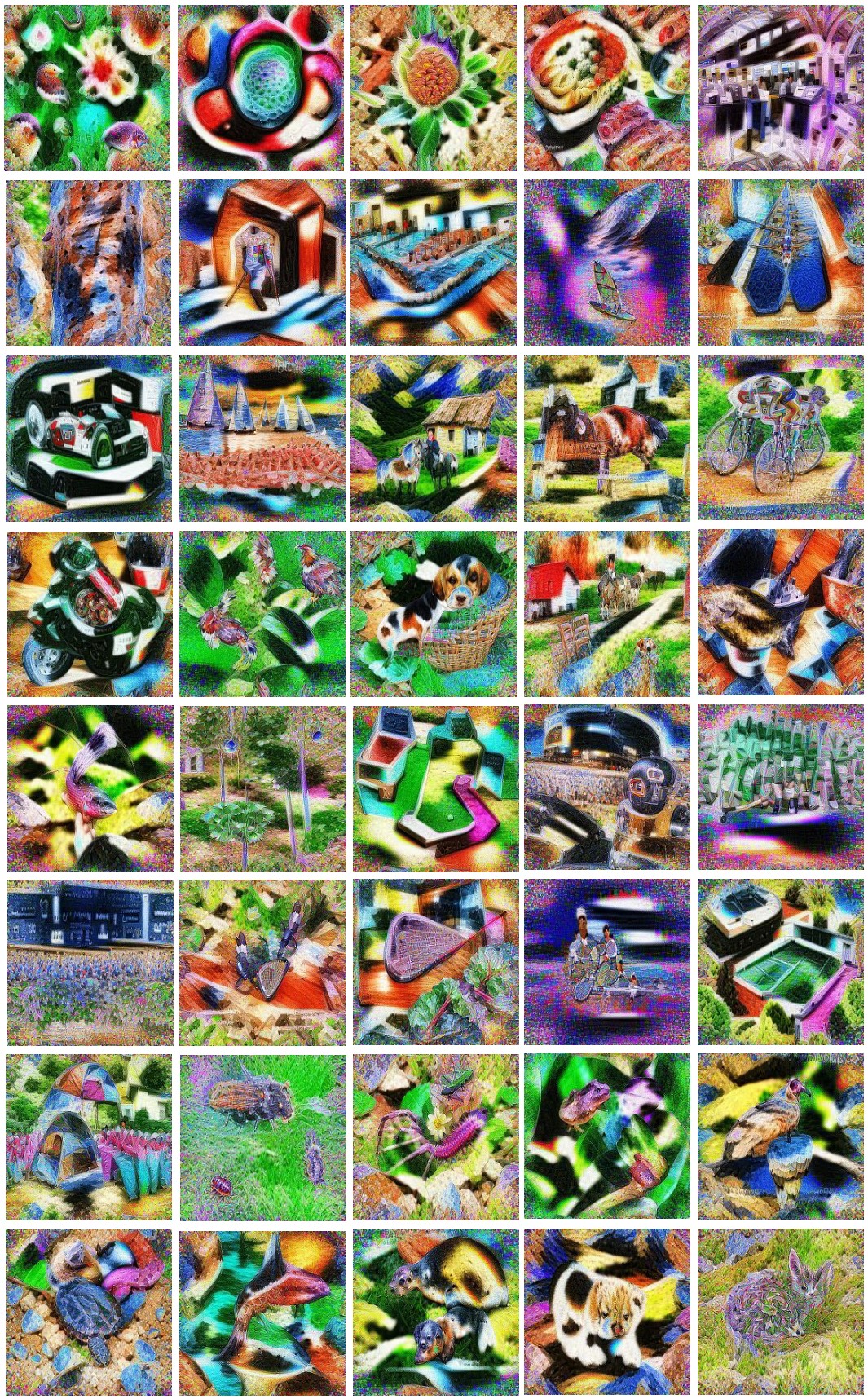

Figure 11: Synthetic ImageNet-21K data distilled from ResNet-50 by CDA.

