# OpenReview forum: "Dataset Distillation in Large Data Era"
_ICLR.cc/2024/Conference — Submitted to ICLR 2024_

### Official Review · Reviewer_tRFy · 2023-10-30

**Soundness:** 3 good
**Presentation:** 3 good
**Contribution:** 3 good
**Rating:** 5
**Confidence:** 4

**Summary:**

This paper propose a dataset distillation method for large-sclae datasets. The authors propose a Curriculum Data Augmentation (CDA) method to improve the baseline Sre2L, achieveing a state-of-the-art perfromance among the existing dataset distillation methods. The atuhors conducted experiments on the large-scale dataset including ImageNet-1k and ImageNet-21k with significant improvement compared with Sr2L.

**Strengths:**

1. The performance is outstanding. The effectiveness of dataset distillation in large-scale datasets has been challenging for the community. This paper proposes a method that generalizes well on ImageNet-1k and ImageNet-21k. Furthermore, the authors provide the distilled dataset for verification, which is convincing.
2. The logic of this paper is rigorously structured, showcasing a well-thought-out approach to the research question. Each argument is methodically developed, drawing on relevant evidence and theoretical frameworks.
3. The experiments of this paper are sufficient. The visualization of the distilled images are good. Figure 2 is good to understand randomresizedCrop.

**Weaknesses:**

1. Although the experiments on ImageNet-1k and ImageNet-21k are convinsing, it would be better to have the experiments on CIFAR10/100 datasets. Because most of the baselines report their results on CIFAR, it will better illustrate the performance improvement of this paper if the authors do so.

2. The authors should highlight the differences (innovation part) of their proposed method compared to the baseline  Sre2L. Section 2 could be organized better to demonstrate the nolvety of this paper.

3. This paper should introduce more about the computational requirements for the experiments on ImageNet-21k. For example, the memory cost, over all training time.

**Questions:**

1. Experiments on CIFAR.

2. The memory cost, over all training time, and GPUs of the experiments on ImageNet-21K.

3. Could this method be mitigated to the dataset such as NLP? The NAS problem in NLP is more urgent than CV.

---

> ### Author Response · Authors · 2023-11-21
> **Response to Reviewer tRFy (Part 1)**
>
> We sincerely thank you for your constructive and insightful comments. We are encouraged by your recognition that our performance is outstanding. The logic of this paper is rigorously structured, showcasing a well-thought-out approach to the research question, and the experiments are sufficient. We would like to address the comments and questions below.
>
> > W1&Q1: Experiments on CIFAR dataset
>
> We provide the comparison on CIFAR-100 dataset among established baselines and our CDA approach. These baseline results are selected from the previous state-of-the-art methods, including Dataset Condensation/Gradient Matching (DC), Differentiable Siamese Augmentation (DSA), Distribution Matching (DM) and Matching Training Trajectories (MTT). Our results with the 800-ep training budget are as follows:
>
> | CIFAR-100 IPC |  DC  | DSA  | DM    | MTT   | Our CDA |
> |-------------|:----------:|:-----------:|:----------:|:-----------:|:-----------:|
> | 1             | 12.8 | 13.9 | 11.4  | **24.3**      | 13.4        |
> | 10            | 25.2 | 32.3 | 29.7  | 40.1   | **49.8**        |
> | 50            |  --  | 42.8 | 43.6  | 47.7   | **64.4**        |
>
> Consider that our approach is a decoupled process of dataset compression followed by a recovery through gradient updating. It is well-suited to large-scale datasets but less so for small IPC values. As anticipated, there is no advantage when IPC value is extremely low, such as IPC = 1. However, when the IPC is increased slightly, our method demonstrates considerable benefits in accuracy over other counterparts.
>
> > W2: The authors should highlight the differences (innovation part) of their proposed method compared to the baseline Sre2L.
>
> Thanks for the valuable suggestion. Both SRe$^2$L and our proposed CDA method utilize local batch mean and variance statistics to match the global statistics of the entire original dataset, synthesizing data by applying gradient updates directly to the image. The impact of such a strategy is that the initial few iterations set the stage for the global structure of the ultimately generated image. However, SRe$^2$L does not capitalize on this characteristic. In contrast, CDA efficiently exploits it by initially employing large crops to capture a more accurate outline of the object. As the process progresses, CDA incrementally reduces the crop size to enhance the finer, local details of the object, significantly elevating the quality of the synthesized data. This strategic method results in a notable ~5% improvement in the final performance, showcasing the innovative contribution of our CDA approach to the dataset distillation task.
>
> > W3&Q2: This paper should introduce more about the computational requirements for the experiments on ImageNet-21k.
>
> For ImageNet-21K, in the squeezing phase, we follow the official scripts in the ImageNet-21K-P dataset (Winter 21 version). It takes 32 hours to squeeze the original training dataset into a ResNet-18 model on 4$\times$ A100 (40G) GPUs. In the recovery phase, we generate ImageNet-21K images with 1 IPC on a single RTX 4090 GPU, taking 11 hours on average. To generate the distilled ImageNet-21K with IPC of 20, it takes about 55 hours on 4$\times$ RTX 4090 GPUs, and the peak GPU memory utilization is 15GB.
>
> For ImageNet-1K, as we have mentioned in Sec. 3.7, there is no additional training cost incurred in our approach comparing to SRe$^2$L. Specifically, we use the off-the-shelf PyTorch's pretrained models as the squeezing model freely. In the recovery phase, to generate the distilled ImageNet-1K with IPC of 50, it takes about 29 hours on a single A100 (40G) GPU and the peak GPU memory utilization is 6.7GB.
>
> In summary of our computational costs and training time, it is evident that, given an equivalent training budget, our results substantially outperform prior approaches. As also illustrated in Figure 1 of the main paper, our CDA method needs only 1K recovery iterations to generate the distilled ImageNet-1K dataset, yet it still manages to significantly outperform the competitive baseline SRe$^2$L, which requires 4K recovery iterations. Therefore, our CDA not only reduces the recovery costs by 75% but also enhances the validation performance.

---

> ### Author Response · Authors · 2023-11-21
> **Response to Reviewer tRFy (Part 2)**
>
> > Q3: Could this method be mitigated to the dataset such as NLP? The NAS problem in NLP is more urgent than CV.
>
> Thanks for this interesting question. Yes, transferring our proposed learning framework from image to language modality is indeed possible, but it requires careful adaptation to account for the fundamental differences between these modalities. The high-level idea of our work involves gradually increasing the complexity or difficulty of synthetic training data, i.e., first synthesizing simple outlines and approximations through gradient updates on the synthetic data, and then tune the details upon it using gradients from harder inputs.
>
> Considering that visual data like images is continuous and spatial, capturing shapes, textures, and colors. Language data is essentially sequential and discrete, consisting of a series of symbols (words, characters). It is highly contextual and the meaning often depends on the sequence and arrangement of these symbols.
>
> What we can do on language is to synthesize by starting with basic textual augmentations such as synonym replacement or slight rephrasing. These changes are relatively minor and do not significantly modify the sentence structure or meaning. Then, we can gradually introduce more complex augmentations that involve restructuring sentences, paraphrasing, or using back-translation (translating a sentence to another language and then back to the original language). On image, we use crop size to control the difficulty of input, on language, adding noise with controlled grammatical errors or word scrambles could be a good practice, and can help the synthetic process to be robust against common errors or variations in language.
>
> We are currently exploring this direction and anticipate having concrete results to share with the community in the near future.

---

### Official Review · Reviewer_Vmep · 2023-10-30

**Soundness:** 3 good
**Presentation:** 3 good
**Contribution:** 2 fair
**Rating:** 5
**Confidence:** 5

**Summary:**

This paper proposes a Curriculum Data Augmentation (CDA) specifically for SRe^2L which is a state-of-the-art dataset distillation method based on the model inversion technique. Specifically, the proposed CDA extends the original random resized crop during the model inversion process of SRe^2L by introducing a scale-varying (from larger to smaller) cropping scheme. Extensive tests demonstrate the effectiveness of the proposed augmentation strategy.

**Strengths:**

S1: The exhibited experimental results demonstrate that the proposed CDA can improve the performance of SRe^2L among various evaluated scenarios.

S2: The paper is well-organized and presents a clear narrative. The experiments showcased are comprehensive and thoughtfully executed.

**Weaknesses:**

W1: It appears that this augmentation strategy is specifically introduced for SRe^2L. Compared to the prior art of data augmentation for dataset distillation (e.g., Data Siamese Augmentation [1]), the universality is considered to be insufficient and thus limits the contribution.

W2: According to the discussion within the original paper of SRe^2L, the primary problem limited to general dataset distillation can be concluded that the data synthesized by SRe^2L are not effective for training models without batch normalization module. It appears that the proposed CDA did not exhibit the capability to mitigate this limitation.

W3: Considering the straightforwardness and intuitiveness of this method, the fact that the authors have not delved deeper into discussing why this CDA strategy can effectively enhance the performance of SRe^2L and the unsatifactory universality of the proposed augementation strategy, it would be considered that this work does not offer sufficient insights for the dataset distillation.

[1] Bo Zhao, Hakan Bilen: Dataset Condensation with Differentiable Siamese Augmentation. ICML 2021: 12674-12685

**Questions:**

Q1: Is this data augmentation strategy also effective for other existing dataset distillation frameworks such as gradient matching or training trajectory matching (MTT)?

Q2: Based on W2, I am wondering if CDA can improve the generalization ability of SRe^2L on models without batch normalization such as ViT?

---

> ### Author Response · Authors · 2023-11-21
> **Response to Reviewer Vmep (Part 1)**
>
> We sincerely thank you for your constructive and insightful feedback. We appreciate your comments that our paper is well-organized and presents a clear narrative. The experiments showcased are comprehensive and thoughtfully executed. We would like to address the concerns and questions below.
>
> > W1: It appears that this augmentation strategy is specifically introduced for SRe$^2$L. Compared to the prior art of data augmentation for dataset distillation (e.g., Data Siamese Augmentation [1]), the universality is considered to be insufficient and thus limits the contribution.
>
> We understand the reviewer's concern and would like to clarify that the universality of our approach is similar to the prior art of data augmentation for dataset distillation, and we further introduce the rationale behind our chosen strategy of the approach to demonstrate our contribution.
>
> DSA (Data Siamese Augmentation) combined several transformations, including color jittering, cropping, cutout, flipping, scaling, and rotation. The performance improvement comes from the ensemble learning with these transformations. In CDA design, we intentionally manipulate crop sizes in a progressive manner throughout different iterations to control the difficulty of the input samples. Provided that the base models are compatible with curriculum learning, our CDA strategy is as broadly applicable as any competing approach like DSA. This implies that whenever curriculum learning yields benefits, our method is equally viable.
>
> Regarding our method's motivation and contribution, our observations indicate that the initial iterations are crucial in establishing the global structure of the ultimate image. The SRe$^2$L approach, however, does not take full advantage of this aspect. In contrast, our CDA method optimally leverages this by starting with larger crops to more accurately delineate the object's contour. As the distillation process unfolds, the proposed gradually decreases the crop size, allowing for fine-grained refinement of the object's detailed features, thereby markedly improving the synthesized data's fidelity. This progression leads to a substantial enhancement, with a roughly 5% increase in the final performance, highlighting the innovative contribution of our proposed CDA on dataset distillation task.
>
> We further provide additional experiments on CIFAR-10 by applying our approach to previous gradient matching (DC) method, where consistent improvement in accuracy is achieved, as shown in the table below.
>
> | CIFAR-10 IPC   | DC+RandomCrop(SRe$^2$L) | DC + CDA (ours) |
> | ------------  | ----------------------- | --------------- |
> | 10           | 44.2                    | **44.4**         |
> | 50          | 52.5                    | **53.0**          |
>
> In summary, our approach enhances the effectiveness of SRe$^2$L on large-scale datasets, achieving significant improvements in both validation accuracy and training efficiency. Our CDA also works well on other frameworks like gradient matching (DC) with consistent improvement.

---

> ### Author Response · Authors · 2023-11-21
> **Response to Reviewer Vmep (Part 2)**
>
> > W2&Q2: According to the discussion within the original paper of SRe^2L, the primary problem limited to general dataset distillation can be concluded that the data synthesized by SRe^2L are not effective for training models without batch normalization module. It appears that the proposed CDA did not exhibit the capability to mitigate this limitation.
>
> We provide additional results in the table below to demonstrate the superior handling of the scenario without BN by our CDA in comparison to the earlier SRe$^2$L approach. We select DeiT-Tiny and VGG models for evaluation under two settings: with and without BN module. As the table indicates, the addition of extra BN layers does not enhance the validation accuracy when training DeiT-Tiny and VGG models, which were originally designed without Batch Norm according to their respective papers. Furthermore, it was observed that models without the Batch Norm module can also be effectively trained on synthetic data to achieve satisfactory performance.
>
> | Method | DeiT-Tiny | VGG-11 | VGG-13 | VGG-16 | VGG-19 |
> |:-------:|:---------:|:------:|:------:|:------:|:------:|
> | Sre$^2$L (ResNet-18 recovery)  |    15.41   | -  |  - |  - |-   |
> | Sre$^2$L (DeiT-Tiny-BN recovery)  |  25.36 |  - | -  |  - | -  |
> | Our CDA w/ BN (ResNet-18 recovery)  |  26.14   | 27.05 | 29.56 | 36.72  | 39.56 |
> | Our CDA w/o BN (ResNet-18 recovery)  |   **31.95**   | **36.99** | **38.60** | **42.28** | **43.30** |
>
> Despite this, it is noteworthy from the table that the validation performance of the DeiT model falls short when compared to that of the conventional ConvNet models. We surmise that the underlying reasons for this discrepancy might include:
>
> 1. *Inherent characteristics of ViT models*. As noted in its original paper, ViT models require a large amount of training data to achieve significant performance. To further validate this, we tested DeiT models on larger distilled datasets with Images Per Class (IPC) values of 100 and 200. The comparison results below show that the accuracy gap between DeiT and ResNet models gradually narrows as the size of the training data increases.
>     | ImageNet-1K IPC | DeiT-Tiny | ResNet-18 |
>     | --------- |:---------:|:---------:|
>     | 50        |   31.95   |   53.5    |
>     | 100       |   46.29   |   58.0    |
>     | 200       |   53.03   |   63.3    |
> 2. *Choosing proper models for recovery*. As discussed in SRe$^2$L, recovering from larger models like ResNet-50 requires more iterations and computational resources to reach similar levels of validation accuracy when compared to smaller counterparts like ResNet-18. Essentially, for a fixed number of recovery iterations, larger models tend to yield lower validation performance compared to smaller models. In our study, Table 15 (Table 18 in our revised version) shows the validation accuracy on synthetic datasets recovered using DenseNet-121, which has more parameters than ResNet-18. Consequently, the validation accuracy of DeiT-Tiny recovered from DenseNet-121 is observed to be lower than that recovered from ResNet-18, as indicated in Table 6 (Table 7 in our revised version).
> 3. *Absence of ViT training tricks and insufficient training budget*. The DeiT-Tiny model was trained using the same codebase and the same hyperparameters as our other validation models. Thus, the training process lacks DeiT (ViT) skills, such as gradient norm clipping, Exponential Moving Average (EMA), etc.

---

> ### Author Response · Authors · 2023-11-21
> **Response to Reviewer Vmep (Part 3)**
>
> > W3: Why this CDA strategy can effectively enhance the performance of SRe$^2$L and the unsatisfactory universality of the proposed augmentation strategy.
>
> We acknowledge and appreciate the reviewer's perspective and comments. However, we respectfully clarify our proposed learning procedure is the result of an in-depth exploration of data synthesis in the context of dataset distillation. We observed that *the initial iterations are crucial for establishing the basic outlines and approximations of the object, with subsequent iterations focusing on refining the details of both the object and its background, as illustrated in Fig. 6 and Fig. 8.* This process underlines the importance of using a larger cropped region in the initial stages. Such an approach lays a comprehensive foundation by encompassing more areas and information, thereby setting the stage for more nuanced and detailed refinements. These refinements are achieved through subsequent gradient updates from harder samples, which progressively focus on smaller, more challenging regions using small crops, enhancing the local details.
>
> Our new angle on data synthesis marks a significant departure from conventional methods. This novel perspective and insights have been pivotal in achieving the highest level of accuracy demonstrated by our work, which surpasses the state-of-the-art performance by ~5%. We believe that this achievement underscores the originality and effectiveness of our method. Furthermore, a straightforward strategy employed elevates the previous state-of-the-art SRe$^2$L on large-scale datasets by an impressive 5%. This achievement should be recognized as a significant merit of our approach, rather than a subject of criticism.
>
> > Q1: Is this data augmentation strategy also effective for other existing dataset distillation frameworks such as gradient matching or training trajectory matching (MTT)?
>
> Thank you for this insightful question regarding the effectiveness of our data augmentation strategy when integrated into other dataset distillation methods.  We carry out experimental integrations of our approach with gradient matching (DC) on the CIFAR-10 dataset, maintaining an IPC of 10. The results show that our CDA method exhibits a modest improvement in performance. This confirms the benefits of embedding curriculum learning within the data augmentation approach.
>
> | CIFAR-10 IPC  | DC+RandomCrop(SRe$^2$L) | DC + CDA (ours) |
> | ------------  |:-----------------------:|:---------------:|
> | 10           | 44.2                    | **44.4**            |
> | 50           | 52.5                    | **53.0**            |
>
> *Responses for Q2* have been incorporated together with W2.

---

> > ### Comment · Reviewer_Vmep · 2023-11-22
> >
> > Thank you for your response. However, from my perspective, the complementary experimental results cannot fully address my concerns because 1. Even disregarding DSA, the CDA's improvement over the random crop (original augmentation method in SRe^2L) for the DC method is considered to be marginal; 2. there is only one additional evaluated method DC was discussed. Considering my weakness 3, I still maintain my current rating.

---

> ### Author Response · Authors · 2023-11-22
> **Clarifications for Reviewer Vmep**
>
> We appreciate the reviewer's further response and concern. We would like to reassure the reviewer that our method represents the most state-of-the-art approach with the highest accuracy to large-scale dataset distillation currently available. It appears there may be some misinterpretations in the reviewer's current observations. We trust that upon reviewing our detailed responses below, the reviewer might reevaluate his/her decision.
>
> 1. The results table referenced by the reviewer in Part 3 pertains to experiments conducted on the CIFAR-10 dataset utilizing the original DC codebase. It should be noted that the marginal improvements observed are attributed to the base model rather than our proposed method. When the DC component is excluded, our methodology yields an enhancement of over 5% on average for SRe$^2$L on datasets such as CIFAR-100, Tiny-ImageNet, ImageNet-1K, and ImageNet-21K. We believe the current criticism from this reviewer lacks fairness, responsibility, and may be misleading to the AC and other reviewers.
>
> 2. We summarize the true improvements by our approach over SRe$^2$L on CIFAR-100, Tiny-ImageNet, ImageNet-1K and ImageNet-21K datasets using the same training setting. We hope that these detailed summaries will restore confidence in our approach for the reviewer.
>
> | CIFAR-100 (IPC) |  DC  | DSA  | DM    | MTT (CVPR'22) | SRe$^2$L (NeurIPS'23) | Our CDA |
> | ------------- |:----:| ---- |:----:|:-------:|:-----------:|:-----------|
> | 10            | 25.2 | 32.3 | 29.7 | 40.1   | --         | &emsp;**49.8$^{\uparrow9.7}$**        |
> | 50            |  --  | 42.8 | 43.6   | 47.7   | 49.4         | &emsp;**64.4$^{\uparrow15.0}$**       |
>
> | Tiny-ImageNet (IPC)        |  DM  | MTT (CVPR'22) | SRe$^2$L (NeurIPS'23)  | Our CDA
> | ---------- |:----:|:-----------:|:-----------:|:-----------|
> | 10         | 12.9 | 23.2  | --   |    &emsp;**43.0$^{\uparrow19.8}$**    |
> | 20         |  --  |  --   | --    |    &emsp;**50.5**    |
> | 50  | 24.1 | 28.0     |  42.5    |    &emsp;**55.5$^{\uparrow13.0}$**    |
>
> | ImageNet-1K (IPC)        |  TESLA (ICML'23)  | SRe$^2$L (NeurIPS'23)    | Our CDA
> | ---------- |:----:|:-----------:|:-----------|
> | 50         | 27.9      | 57.6 |   &emsp;  **61.6$^{\uparrow4.0}$**    |
> | 100         | --      | 62.8 |   &emsp;  **65.9$^{\uparrow3.1}$**    |
>
> | ImageNet-21K (IPC)         | SRe$^2$L (NeurIPS'23)    | Our CDA
> | ---------- |:----:|:-----------:|
> | 10         | 27.3  |   &emsp; **34.2$^{\uparrow6.9}$**    |
> | 20        |  31.8   |  &emsp; **36.1$^{\uparrow4.3}$**    |
>
> 3. In response to the comment regarding "there is only one additional evaluated method DC was discussed", we wish to respectfully clarify that the rebuttal period is constrained by time. Given this limited timeframe, we have prioritized addressing all of the reviewers' concerns and carrying out the necessary additional experiments. We believed that including one demonstrative experiment would suffice for the purpose of addressing this concern. Currently, with less than a day remaining for the rebuttal, it is not feasible for us to complete the MTT due to its considerable demand on time for training. However, since ICLR is open-reviewed (thanks for the good mechanism of reviewing), we are able to and we are committed to finishing this task post-rebuttal for this reviewer, we will post the results both here and in the revised paper.
>
> Most importantly, we wish to express our perspective that **significant advancements in this field warrant impartial evaluation rather than intentional diminishment**.

---

### Official Review · Reviewer_7cBa · 2023-10-30

**Soundness:** 3 good
**Presentation:** 3 good
**Contribution:** 2 fair
**Rating:** 5
**Confidence:** 2

**Summary:**

The paper introduces a dataset distillation model that employs a random cropping strategy for synthetic images, progressing from coarse to fine using various sampling scale hyperparameters. This cropping approach integrates the concept of Curriculum Learning. Notably, the model's efficiency and capability to handle large-scale dataset distillation are achieved without relying on complex gradient matching or trajectory matching techniques. Experimental results demonstrate that the model surpasses state-of-the-art models, including SRe2L, in multiple ImageNet subsets, including a large-scale ImageNet-21K dataset.

**Strengths:**

1. The first effort to apply a larger-scale dataset, i.e., ImageNet-21K in dataset distillation. It will amplify attention on this method for larger datasets, enhancing comprehension of its advantages and real-world challenges.
2. The model is simple yet it outperforms the existing method SRe2L by a margin.

**Weaknesses:**

1. The technical innovation is rather limited; the approach is more of a simple data augmentation trick. Although the concept of Curriculum Learning is incorporated, there's a lack of further exploration regarding the rationale or underlying principles for its application in the context of dataset distillation.
2. The experimental assessment is insufficient, lacking comparisons with a broader range of state-of-the-art models such as KIP, TM, and DSA, across various settings, including different architectures, IPC values, and image types.

**Questions:**

1. Regarding Eq. (3), could you explain why the objective is defined based on the training data (D) rather than on the distribution of real data?
2. The specific details on how to leverage the statistics in Batch Normalization (BN) have not been provided.
3. For the random crop: (a) Why is only the min_crop varied and not both min_crop and max_crop? (b) Should we always set the parameter \beta_u to 1 to maintain the original resolution intact?
4. In Algorithm 1, how does the ReverseRandomResizedCrop function operate? Does it directly restore and scale a crop, or is it applied as a patch to the original image? Additionally, aside from preserving the same resolution, does it serve any other purpose in the overall process?
5. The settings for reverse curriculum learning are not entirely clear. Could you please specify the actual values of \beta_l and \beta_u that are used during training?
6. The experiments conducted in the study is not sufficient: (a) It would be valuable to include cross-architecture performance evaluations. Instead of exclusively training on the ResNet family (e.g. Table 6 and 7), consider testing the approach with a broader range of models such as LeNet, VGG, MLP, etc. (b) Expanding the evaluation datasets beyond ImageNet to include other datasets like SVHN, MNIST, etc., would provide a more comprehensive assessment of the method's performance. (c) More baselines should be compared, such as Gradient Matching, Differentiable Siamese Augmentation, Distribution Matching, KIP, and Training Trajectory Matching (d) Considering a wider range of IPC settings, especially with smaller IPC values like 1, 10, 20, would offer insights into the method's performance across various scenarios.
7. It would be better to include a time and efficiency comparison when training on massive datasets like Image-21k. Providing insights into the computational costs and time requirements for training on such datasets for DD methods.

Minor comments:
1. Please provide the full name of an abbreviation, for example, IPC, which stands for image per class.
2. CRL in page 6 -> RCL

---

> ### Author Response · Authors · 2023-11-21
> **Response to Reviewer 7cBa (Part 1)**
>
> We sincerely thank you for your constructive and detailed comments. We appreciate your recognition of our work as the first effort to apply a larger-scale ImageNet-21K in dataset distillation, and our model is simple yet can outperform the existing state-of-the-art method by a margin. We would like to address each of your comments and questions individually.
>
> > W1: More of a simple data augmentation trick. Although the concept of Curriculum Learning is incorporated, there’s a lack of further exploration regarding the rationale or underlying principles for its application in the context of dataset distillation.
>
> We acknowledge and appreciate the reviewer's perspective and feedback. However, we respectfully disagree with the comment suggesting that our approach is merely a simple data augmentation trick. Our proposed learning procedure is the result of an in-depth exploration of data synthesis in the context of dataset distillation. We observed that *the initial iterations are crucial for establishing the basic outlines and approximations of the object, with subsequent iterations focusing on refining the details of both the object and its background, as illustrated in Fig. 6 and Fig. 8.* This process underlines the importance of using a larger cropped region in the initial stages. Such an approach lays a comprehensive foundation by encompassing more areas and information, thereby setting the stage for more nuanced and detailed refinements. These refinements are achieved through subsequent gradient updates from harder samples, which progressively focus on smaller, more challenging regions using small crops, enhancing the local details.
>
> Our new angle on data synthesis marks a significant departure from conventional methods. This novel perspective and exploration have been pivotal in achieving the highest level of accuracy demonstrated by our work, which surpasses the state-of-the-art performance by ~5%. We believe that this achievement underscores the originality and effectiveness of our method, distinguishing it from simpler data augmentation techniques. Furthermore, a straightforward strategy employed elevates the previous state-of-the-art SRe$^2$L (NeurIPS'23) on large-scale datasets by an impressive 5%. This achievement should be recognized as a significant merit of our approach, rather than a subject of criticism.

---

> ### Author Response · Authors · 2023-11-21
> **Response to Reviewer 7cBa (Part 2)**
>
> > W2: The experimental assessment is insufficient, lacking comparisons with a broader range of state-of-the-art models such as KIP, TM, and DSA, across various settings, including different architectures, IPC values, and image types.
>
> Thank you for your valuable feedback regarding the scope of our experimental assessment. In response, we have conducted additional experiments (trained with  800-ep budget) and incorporated a detailed comparison with state-of-the-art methods on CIFAR-100 and Tiny ImageNet datasets across various IPC settings, shown in the following two tables.
>
> | CIFAR-100 IPC |  DC  | DSA  | DM   | KIP  | MTT | SRe$^2$L  | Our CDA |
> | ------------- |:----:| ---- |:----:|:----:|:-------:|:-----------:|:-----------:|
> | 1             | 12.8 | 13.9 | 11.4 | **34.9** | 24.3   | -          | 13.4        |
> | 10            | 25.2 | 32.3 | 29.7 | 49.5 | 40.1   | -         | **49.8**        |
> | 50            |  --  | 42.8 | 43.6 | --   | 47.7   | 49.4         | **64.4**       |
>
> | Tiny-ImageNet IPC        |  DM  | MTT | SRe$^2$L   | Our CDA
> | ---------- |:----:|:-----------:|:-----------:|:-----------:|
> | 1          | 3.9  | **8.8**      |  --   |    3.29 $\pm$ 0.26     |
> | 10         | 12.9 | 23.2  | --   |    **43.04 $\pm$ 0.26**    |
> | 20         |  --  |  --   | --    |    **50.46 $\pm$ 0.14**    |
> | 50  | 24.1 | 28.0     |  42.5    |    **55.50 $\pm$ 0.18**    |
>
> We further expand the scope of validation models on ImageNet-1K, including SqueezeNet, MobileNet, EfficientNet, MNASNet, ShuffleNet, ResMLP, AlexNet, DeiT-Base, and VGG family models.
>
> | **Model**    | **SqueezeNet** | **MobileNet** | **EfficientNet** | **MNASNet** | **ShuffleNet** | **ResMLP** |
> |--------------|:--------------:|:-------------:|:----------------:|:-----------:|:--------------:|:----------:|
> | #Params (M)  |       1.2      |      3.5      |        5.3       |     6.3     |       7.4      |    30.0    |
> | accuracy (%) |     19.70     |     49.76    |       55.10       |    55.66   |     54.69     |   54.18   |
> | **Model**    |   **AlexNet**  | **DeiT-Base** |    **VGG-11**    |  **VGG-13** |   **VGG-16**   | **VGG-19** |
> | #Params (M)  |      61.1      |      86.6     |       132.9      |    133.0    |      138.4     |    143.7   |
> | accuracy (%) |     14.60     |     30.27    |      36.99      |    38.60   |     42.28     |   43.30   |
>
> Consider that our approach is a decoupled process of dataset compression followed by recovery through gradient updating. It is well-suited to large-scale datasets but less so for small IPC values. As anticipated, there is no advantage when IPC value is extremely low, such as IPC = 1. However, when the IPC is increased slightly, our method demonstrates considerable benefits in accuracy over other counterparts.
>
> In summary, our CDA also achieves the state-of-the-art performance on CIFAR and Tiny-ImageNet under the IPC settings of 10, 20, and 50. On ImageNet-1K, our CDA surpasses SRe$^2$L, achieving superior performance in both validation accuracy and training efficiency. Our distilled dataset has been evaluated on a broad range of models to demonstrate its significant cross-model generalization. More details and discussions are included in our responses to Q6(a)(b)\(c\)(d).
>
> > Q1: Regarding Eq. (3), could you explain why the objective is defined based on the training data (D) rather than on the distribution of real data?
>
> In Eq. 3, *sup* is the supremum representing the least upper bound of a set. In this context, it means we are looking for the maximum difference between the losses of two models (original data trained model $\phi_{\boldsymbol{\theta}\_{\mathcal{D}\_o}}$ and distilled data trained model $\phi_{\boldsymbol{\theta}_{\mathcal{D}_d}}$). The supremum ensures that we consider the worst-case scenario. Specifically, the whole Eq. 3 is describing an optimization process where the goal is to minimize the maximum discrepancy between the loss incurred by a model trained on the original data $\mathcal{D}_o$ and a model trained on the distilled $\mathcal{D}_d$, across all input-label pairs sampled from $\mathcal{D}_o$ ($(\boldsymbol{x}, \boldsymbol{y}) \sim \mathcal{D}_o$), i.e., evaluating both these two models on the real original validation set to make sure the model trained on the distilled data has the best generalization accuracy when testing on the real val data. This equation describes the optimization goal of the dataset distillation task.

---

> ### Author Response · Authors · 2023-11-21
> **Response to Reviewer 7cBa (Part 3)**
>
> > Q2: The specific details on how to leverage the statistics in Batch Normalization (BN) have not been provided.
>
> Thank you for raising this question to us. The detailed loss formulation of utilizing statistics in BN is provided as follows, which has been included in our revision:
>
> \begin{equation}
> \begin{aligned}
> \mathcal{R}_{\mathrm{reg}}\left(\boldsymbol{x}^{\prime}\right) & =\sum_k\left\|\mu_k\left(\boldsymbol{x}^{\prime}\right)-\mathbb{E}\left(\mu_k \mid \mathcal{D}_o\right)\right\|_2+\sum_k\left\|\sigma_l^2\left(\boldsymbol{x}^{\prime}\right)-\mathbb{E}\left(\sigma_k^2 \mid \mathcal{D}_o\right)\right\|_2 \\\\
> & \approx \sum_k\left\|\mu_k\left(\boldsymbol{x}^{\prime}\right)-\mathbf{B} \mathbf{N}_k^{\mathrm{RM}}\right\|_2+\sum_k\left\|\sigma_k^2\left(\boldsymbol{x}^{\prime}\right)-\mathbf{B} \mathbf{N}_k^{\mathrm{RV}}\right\|_2
> \end{aligned}
> \end{equation}
>
> where $k$ is the index of BN layer, $\mu_k\left(\boldsymbol{x}^{\prime}\right)$ and $\sigma_k^2\left(\boldsymbol{x}^{\prime}\right)$ are the channel-wise mean and variance in current batch data. $\mathbf{B N}_k^{\mathrm{RM}}$ and $\mathbf{B N}_k^{\mathrm{RV}}$ are mean and variance in the pre-trained model at $k$-th BN layer, which are globally counted.
>
> In our procedure, we first crop and resize images on input batch data in a curriculum learning way and feed them into the model. At each BN layer, we calculate the current feature map's channel-wise mean and variance in a batch. Then, these statistics are compared with the mean and variance values stored in the corresponding BN layer of the pre-trained model. This comparison is incorporated into the final loss function as $R_{reg}$ in Eq. (6) of our paper.
>
> > Q3: For the random crop: (a) Why is only the min_crop varied and not both min_crop and max_crop? (b) Should we always set the parameter \beta_u to 1 to maintain the original resolution intact?
>
> Thanks for your detailed questions and we clarify our random crop strategy as follows:
>
> (a) In CL/RCL experiments, max_crop of 1 is not varied to ensure that the entire image has a probability of being cropped and updated simultaneously throughout the whole process. However, in CTL experiments, max_crop is varied to control the level of easy type.
>
> (b) Yes, based on the ablation of our CL/RCL experiments in Table 2 of the paper, we maintain the max_crop ($\beta_u$) parameter at constant 1.0 to guarantee that the global information on the entire image has a probability of being updated simultaneously.
>
> > Q4: In Algorithm 1, how does the ReverseRandomResizedCrop function operate? Does it directly restore and scale a crop, or is it applied as a patch to the original image? Additionally, aside from preserving the same resolution, does it serve any other purpose in the overall process?
>
> Thank you for your careful scrutiny of the details on the operation of this crop function. ReverseRandomResizedCrop operation is the inverse process of RandomResizedCrop. It involves inversely resizing the model's input image ${x}^{\prime}_\mathcal{T}$ (224 $\times$ 224) back to its original crop size, and then replacing it in the crop location of the synthetic images $x_s$. The combined use of RandomResizedCrop and ReverseRandomResizedCrop functions ensures that, in the current iteration, only the cropped area of the entire image is updated.
>
> > Q5: The settings for reverse curriculum learning. Could you please specify the actual values of \beta_l and \beta_u that are used during training?
>
> Sec. 3.1 and 3.3 detail the configurations of RCL experiments in the Reverse Curriculum Learning paragraph. As we mentioned in Sec. 3.1, the $\beta_l$ and $\beta_u$ refer to the default lower and upper bounds in PyTorch's RandomResizeCrop, which are set to 0.08 and 1.0, respectively. During training, the current lower bounds $\alpha_l$ and upper bounds $\alpha_u$ are dynamically adjusted by augmentation style and difficulty scheduler, as shown in Alg. 1 and Fig. 4.

---

> ### Author Response · Authors · 2023-11-21
> **Response to Reviewer 7cBa (Part 4)**
>
> > Q6 (a): It would be valuable to include cross-architecture performance evaluations. Instead of exclusively training on the ResNet family (e.g. Table 6 and 7), consider testing the approach with a broader range of models such as LeNet, VGG, MLP, etc.
>
> Thank you for suggesting including more different architecture models to evaluate the cross-architecture performance. As suggested by the reviewer, we have conducted additional validation experiments on a broad range of models, including SqueezeNet, MobileNet, EfficientNet, MNASNet, ShuffleNet, ResMLP, AlexNet, DeiT-Base, and VGG family models. These validation models are selected from a wide variety of architectures, encompassing a vast range of parameters.
>
> | **Model**    | **SqueezeNet** | **MobileNet** | **EfficientNet** | **MNASNet** | **ShuffleNet** | **ResMLP** |
> |--------------|:--------------:|:-------------:|:----------------:|:-----------:|:--------------:|:----------:|
> | #Params (M)  |       1.2      |      3.5      |        5.3       |     6.3     |       7.4      |    30.0    |
> | accuracy (%) |      19.70     |     49.76     |       55.10      |    55.66    |      54.69     |    54.18   |
> | **Model**    |   **AlexNet**  | **DeiT-Base** |    **VGG-11**    |  **VGG-13** |   **VGG-16**   | **VGG-19** |
> | #Params (M)  |      61.1      |      86.6     |       132.9      |    133.0    |      138.4     |    143.7   |
> | accuracy (%) |      14.60     |     30.27     |       36.99      |    38.60    |      42.28     |    43.30   |
>
> In the upper group of the table, the selected models are relatively small and efficient. There is a trend that its validation performance improves as the number of model parameters increases. In the lower group, we validated earlier models AlexNet and VGG. These models also show a trend of performance improvement with increasing size, but due to the simplicity of early model architectures, such as the absence of residual connections, their performance is inferior compared to more recent models. Additionally, we evaluated our distilled dataset on ResMLP, which is based on MLPs, and the DeiT-Base model, which is based on transformers. In summary, the distilled dataset created using our CDA method demonstrates strong validation performance across a wide range of models, considering both architecture diversity and parameter size.
>
> > Q6 (b): Expanding the evaluation datasets beyond ImageNet to include other datasets, would provide a more comprehensive assessment of the method's performance.
>
> Thank you for your insightful suggestion regarding conducting additional experiments on small-scale datasets beyond ImageNet. Consider that SVHN and MNIST are a little bit outdated and not so popular now in the dataset distillation community. Here, we conduct additional experiments on a small dataset CIFAR-100 with 32 $\times$ 32 resolution.
>
> In detail, according to the small scale of CIFAR images, which is 32 $\times$ 32 resolution, the default lower bound $\beta_l$ needs to be raised from 0.08 (ImageNet setting) to a higher reasonable value in order to avoid the training inefficiency caused by extremely small cropped areas with little information. Thus, we conducted the ablation to select the optimal value for the default lower bound $\beta_l$ in RandomResizedCrop operations. In the table below, we choose 0.4 as the default lower bound $\beta_l$ in Alg. 1.
>
> | CDA lower bound      | 0.08 |  0.2  |  0.4 |  0.6  |  0.8  |  1.0  |
> |----------------------|:----:|:-----:|:----:|:-----:|:-----:|:-----:|
> | accuracy (800ep) (%) | 58.5 | 62.14 | **64.4** | 63.36 | 61.65 | 54.43 |
>
> To further improve the validation accuracy, we adopt a small batch size value of 8 and extend the training budgets in the following validation stage, which aligns with the strong training recipe design discussed in Appendix D of our paper. In the table below, we observe a trend that the validation model's accuracy exhibits a significant improvement along with the extension of training budgets. The experimental results demonstrate the effectiveness of our method on small datasets beyond ImageNet.
>
> | CIFAR-100 IPC | CDA (100ep) | CDA (200ep) | CDA (400ep) | CDA (800ep) |
> | ------------- |:-----------:|:-----------:|:-----------:|:-----------:|
> | 1             |     7.1     |     8.2     |    10.2     |    13.4     |
> | 10            |    25.0     |    34.9     |    44.5     |    49.8     |
> | 50            |    48.9     |    56.6     |    60.4     |    64.4     |

---

> ### Author Response · Authors · 2023-11-21
> **Response to Reviewer 7cBa (Part 5)**
>
> > Q6 \(c\): More baselines should be compared, such as Gradient Matching, Differentiable Siamese Augmentation, Distribution Matching, KIP, and Training Trajectory Matching
>
> Thanks for your kind suggestion about including more previous baselines. We have extended the CIFAR-100 results in Q6 (b) with the state-of-the-art counterparts, including Dataset Condensation/Gradient Matching (DC), Differentiable Siamese Augmentation (DSA), Kernel Inducing Points (KIP), Distribution Matching (DM) and Matching Training Trajectories (MTT). As presented in the table below, we notice that under the IPC setting of 10 and 50, our CDA's validation accuracy (800-ep training budget) outperforms the competitive baselines with significant margins from 0.3\% to 16.7\%. Our results have the potential to be further improved when training with more budgets. Overall, our CDA method is also applicable for the dataset distillation task on small-scale datasets like CIFAR.
>
> | CIFAR-100 IPC |  DC  | DSA  | DM   | KIP  | MTT   | Our CDA |
> | ------------- |:----:|:----:|:----:|:----:|:----:|:-----------:|
> | 1             | 12.8 | 13.9 | 11.4 | 34.9 | 24.3   | 13.4        |
> | 10            | 25.2 | 32.3 | 29.7 | 49.5 | 40.1  | 49.8        |
> | 50            |  --  | 42.8 | 43.6 | --   | 47.7  | 64.4        |
>
> > Q6 (d): Considering a wider range of IPC settings, especially with smaller IPC values like 1, 10, 20, would offer insights into the method's performance across various scenarios.
>
> Thank you for your suggestion to extend our experimental analysis to the dataset with smaller compression ratios. In the Table presented in response to Q6\(c\), we discussed the distilled CIFAR datasets with smaller compression ratios. In addition, we conducted extensive experiments on our distilled Tiny-ImageNet dataset with smaller IPC values of 1, 10, and 20. We have generated distilled datasets with small IPC by sampling from a large distilled Tiny-ImageNet dataset with an IPC of 200. The validation experiments were conducted 5 times for each IPC setting, allowing us to present average validation results for the distilled datasets with smaller IPC. Furthermore, we have incorporated results from previous methods, Distribution Matching (DM) and Matching Training Trajectories (MTT), as baselines.
>
> | Tiny-ImageNet IPC        |  DM  | MTT   | CDA (200ep) | CDA (400ep) | CDA (800ep) |
> | ---------- |:----:|:-----------:|:-----------:|:-----------:|:-----------:|
> | 1          | 3.9  | 8.8      |     2.38 $\pm$ 0.08     |    2.82 $\pm$ 0.06     |    3.29 $\pm$ 0.26     |
> | 10         | 12.9 | 23.2    |    30.41 $\pm$ 1.53    |    37.41 $\pm$ 0.02   |    43.04 $\pm$ 0.26    |
> | 20         |  --  |  --     |    43.93 $\pm$ 0.20    |    47.76 $\pm$ 0.19    |    50.46 $\pm$ 0.14    |
> | 50  | 24.1 | 28.0     |    50.26 $\pm$ 0.09     |    51.52 $\pm$ 0.17    |    55.50 $\pm$ 0.18    |
>
> Consider that our approach is a decoupled process of dataset compression followed by recovery through gradient updating. It is well-suited to large-scale datasets but less so for small IPC values. As anticipated, there is no advantage when IPC value is extremely low, such as IPC = 1. However, when the IPC is increased slightly, our method demonstrates considerable benefits on accuracy over other counterparts. Furthermore, we emphasize that our approach yields substantial improvements when afforded a larger training budget, i.e., more training epochs.

---

> ### Author Response · Authors · 2023-11-21
> **Response to Reviewer 7cBa (Part 6)**
>
> > Q7: It would be better to include a time and efficiency comparison when training on massive datasets like Image-21k. Providing insights into the computational costs and time requirements for training on such datasets for DD methods.
>
> For ImageNet-21K, in the squeezing phase, we follow the official scripts in the ImageNet-21K-P dataset (Winter 21 version). It takes 32 hours to squeeze the original training dataset into a ResNet-18 model on 4$\times$ A100 (40G) GPUs. In the recovery phase, we generate ImageNet-21K images with 1 IPC on a single RTX 4090 GPU, taking 11 hours on average. To generate the distilled ImageNet-21K with IPC of 20, it takes about 55 hours on 4$\times$ RTX 4090 GPUs, and the peak GPU memory utilization is 15GB.
>
> For ImageNet-1K, as we have mentioned in Sec. 3.7, there is no additional training cost incurred in our approach comparing to SRe$^2$L. Specifically, we use the off-the-shelf PyTorch's pretrained models as the squeezing model freely. In the recovery phase, to generate the distilled ImageNet-1K with IPC of 50, it takes about 29 hours on a single A100 (40G) GPU and the peak GPU memory utilization is 6.7GB.
>
> For the insights into the computational costs and time and derive a summary of them, it is evident that, given an equivalent training budget, our results substantially outperform prior approaches. As illustrated in Figure 1 of the main paper, our CDA method needs only 1K recovery iterations to generate the distilled ImageNet-1K dataset, yet it still manages to significantly outperform the competitive baseline SRe$^2$L, which requires 4K recovery iterations. Therefore, our CDA not only reduces the recovery costs by 75% but also enhances the validation performance.
>
> It's noteworthy that the trend in dataset distillation is to compress large datasets, like ImageNet, which can greatly reduce the training cost from several days to hours. On the contrary, it only takes 10 minutes to train a ResNet-18 model with over 90\% Val accuracy on the original CIFAR-10, and 20 seconds to train a ConvNet model for merely an epoch on the original MNIST to achieve over 98\% accuracy. Therefore, low-cost model training on small-scale and simple datasets does not provide a promising guide for real application in the field of data distillation.
>
> >Minor comments of typos.
>
> Thanks for pointing them out. We have corrected them in our revised paper, and please check them out.

---

### Official Review · Reviewer_tmMV · 2023-11-01

**Soundness:** 2 fair
**Presentation:** 3 good
**Contribution:** 3 good
**Rating:** 5
**Confidence:** 5

**Summary:**

The paper introduces a curriculum learning framework for dataset distillation. It presents a strategic learning approach during the data recovery and synthesis phase, where image crops are adaptively adjusted according to the complexity of regions. The study investigates three learning paradigms for data synthesis: standard curriculum learning, reverse curriculum learning, and constant learning. Extensive experiments showcase the promising and superior results achieved by the proposed method.

**Strengths:**

1. The introduction of this article is well-crafted. Algorithm 1 efficiently conveys the author's method to the readers.

2. The design of the CDA framework is innovative, delivering good results on both ImageNet-1K and ImageNet-21K.

3. The inclusion of coarse data synthesis not only stabilizes the training process but also improves the model's ability to generalize and reduces the risk of overfitting.

**Weaknesses:**

1. In Table 2, the authors compared the distillation performance of their proposed method with SRe2L in Tiny ImageNet, ImageNet-1K, and ImageNet-21K datasets. However, I am curious about the performance when randomly sampling an equal number of images from the source dataset, which was not explicitly shown in the table (e.g., randomly selecting 200 images from ImageNet-1K and training on resnet-18).

2. The validation accuracy for DeiT-Tiny in Table 15 appears to be subpar. What do you think could be the reason for this result?

3. Given that other dataset distillation approaches have shown effectiveness at higher distillation ratios, could the authors provide experimental results for this method on the Tiny ImageNet dataset with smaller compression ratios (e.g., IPC=1 and IPC=10)?

**Questions:**

See my comments in weakness section.

---

> ### Author Response · Authors · 2023-11-21
> **Response to Reviewer tmMV (Part 1)**
>
> We sincerely thank you for your constructive and insightful comments. We are encouraged that you recognize our design is innovative, delivering good results on both ImageNet-1K and ImageNet-21K. We would like to address the comments and questions below.
>
> >W1: I am curious about the performance when randomly sampling an equal number of images from the source dataset, which was not explicitly shown in the table (e.g., randomly selecting 200 images from ImageNet-1K and training on resnet-18).
>
> Thank you for your insightful comment regarding random baseline results on the original dataset with a similar IPC setting. Following the suggestion, we conducted additional experiments to randomly sample images from ImageNet-1K training data under several IPC settings, and train ResNet models on these subsets using official PyTorch recipes. The Top-1 validation accuracy comparisons are provided below.
>
> | ImageNet-1K IPC    | ResNet-18 |      | ResNet-50 |      | ResNet-101 |      |
> |-----|:---------:|:----:|:---------:|:----:|:----------:|:----:|
> |  |   Random  |  CDA |   Random  |  CDA |   Random   |  CDA |
> | 50  |   27.47   | **53.5** |   26.05   | **61.3** |    26.96   | **61.6** |
> | 100 |   41.23   | **58.0** |   42.93   | **65.1** |    42.84   | **65.9** |
> | 200 |   52.07   | **63.3** |   55.26   | **67.6** |    57.41   | **68.4** |
>
> Under the same IPC settings, our CDA demonstrated an improvement of at least 10\% over the results on the original dataset. Moreover, as the IPC gradually decreases, the performance advantage exhibited by our distilled data becomes more evident. This aligns with the principle and goal of dataset distillation.
>
> >W2: The validation accuracy for DeiT-Tiny in Table 15 appears to be subpar. What do you think could be the reason for this result?
>
> Thank you for your careful scrutiny of the validation accuracy for DeiT-Tiny (referenced as Table 19 in our revised paper). Your detailed observation is highly valued. Upon a thorough review of our method and experimental procedures, we have pinpointed several potential factors that might have led to the less-than-expected results for DeiT-Tiny:
>
> 1. *Inherent characteristics of ViT models*. As noted in its original paper, ViT models require a large amount of training data to achieve significant performance. To further validate this, we tested DeiT models on larger distilled datasets with Images Per Class (IPC) values of 100 and 200. The comparison results below show that the accuracy gap between DeiT and ResNet models gradually narrows as the size of the training data increases.
>     | ImageNet-1K IPC | DeiT-Tiny | ResNet-18 |
>     | --------- |:---------:|:---------:|
>     | 50        |   31.95   |   53.5    |
>     | 100       |   46.29   |   58.0    |
>     | 200       |   53.03   |   63.3    |
> 2. *Choosing proper models for recovery*. As discussed in SRe$^2$L, recovering from larger models like ResNet-50 requires more iterations and computational resources to reach similar levels of validation accuracy when compared to smaller counterparts like ResNet-18. Essentially, for a fixed number of recovery iterations, larger models tend to yield lower validation performance compared to smaller models. In our study, Table 15 (Table 18 in our revised version) shows the validation accuracy on synthetic datasets recovered using DenseNet-121, which has more parameters than ResNet-18. Consequently, the validation accuracy of DeiT-Tiny recovered from DenseNet-121 is observed to be lower than that recovered from ResNet-18, as indicated in Table 6 (Table 7 in our revised version).
> 3. *Absence of ViT training tricks and insufficient training budget*. The DeiT-Tiny model was trained using the same codebase and the same hyperparameters as our other validation models. Thus, the training process lacks DeiT (ViT) skills, such as gradient norm clipping, Exponential Moving Average (EMA), etc.

---

> ### Author Response · Authors · 2023-11-21
> **Response to Reviewer tmMV (Part 2)**
>
> >W3: Could the authors provide experimental results for this method on the Tiny ImageNet dataset with smaller compression ratios (e.g., IPC=1 and IPC=10)?
>
> Thank you for your insightful suggestion to extend our experimental analysis. In response, we have created condensed versions of the Tiny-ImageNet dataset with lower IPCs. The validation experiments were conducted 5 times for each IPC setting, allowing us to present average validation results for the distilled datasets with smaller IPC. Furthermore, we have incorporated results from previous methods, Distribution Matching (DM), Matching Training Trajectories (MTT) and Batch Norm Matching (SRe$^2$L), as baselines.
>
> | Tiny-ImageNet IPC        |  DM  | MTT  | SRe$^2$L   | CDA (200ep) | CDA (400ep) | CDA (800ep) |
> | ---------- |:----:|:-----------:|:-----------:|:-----------:|:-----------:|:-----------:|
> | 1          | 3.9  | **8.8**    |  --  |     2.38 $\pm$ 0.08     |    2.82 $\pm$ 0.06     |    3.29 $\pm$ 0.26     |
> | 10         | 12.9 | 23.2  |  --  |    **30.41 $\pm$ 1.53**    |    37.41 $\pm$ 0.02   |    43.04 $\pm$ 0.26    |
> | 20         |  --  |  --   | --  |    **43.93 $\pm$ 0.20**    |    47.76 $\pm$ 0.19    |    50.46 $\pm$ 0.14    |
> | 50  | 24.1 | 28.0   |  42.5  |    **50.26 $\pm$ 0.09**     |    51.52 $\pm$ 0.17    |    55.50 $\pm$ 0.18    |
>
> Consider that our approach is a decoupled process of dataset compression followed by a recovery through gradient updating. It is well-suited to large-scale datasets but less so for small IPC values. As anticipated, there is no advantage when the IPC value is extremely low, such as IPC = 1. However, when the IPC is increased slightly, our method demonstrates considerable benefits in accuracy over other counterparts. Furthermore, we emphasize that our approach yields substantial improvements when afforded a larger training budget, i.e., more training epochs.

---

### Author Response · Authors · 2023-11-21
**Summary of Rebuttal&Revision**

We would like to express our gratitude for the valuable feedback and insights from all reviewers, which have significantly contributed to the improvement and refinement of our submission. We kindly invite you to review our author rebuttal and revised paper so that we may address any further questions you may have, or clarify any points that remain unclear. In summary, our rebuttal mainly includes the following:

- We provide the comparison between our CDA and random sampling of the same amount of images from the original dataset. (Reviewer tmMV)
- We analyze the reasons for subpar validation accuracy on DeiT-Tiny. (Reviewer tmMV, Vmep)
- We expand our experimental results to include scenarios with limited Images Per Class (IPC) on CIFAR and Tiny-ImageNet datasets, and comparison with previous state-of-the-art methods. (Reviewer tmMV, 7cBa, Vmep, tRFy)
- We explain the underlying principles and motivations of our CDA approach, emphasizing the novelty and impact of our work. (7cBa, Vmep, tRFy)
- We detail the computational demands and consumptions, and provide a comparative analysis for both ImageNet-1K and ImageNet-21K datasets. (7cBa, tRFy)
- We elaborate on the details about Batch Normalization matching and RandomResizedCrop mechanism under CL, RCL and CTL. (7cBa)
- We evaluate the cross-model generalization ability of our distilled dataset on a wide range of models. (7cBa, Vmep)
- We discuss the feasibility of applying our dataset distillation framework, originally developed for image modality, to the language domain. (tRFy)

We hope our responses can adequately resolve your concerns. We have integrated these points into our revised paper, and we sincerely appreciate your valuable feedback.

---

### Meta-Review · Area_Chair_J7pM · 2023-12-04

**Metareview:**

This paper proposes a Curriculum Data Augmentation (CDA) method for dataset distillation, achineving the SOTA performance on the ImageNet datasets. The authors incorporate the concept of Curriculum Learning to design the augmentation method CDA, addressing the cropping issue of the baseline SRe$^2$L. After a thorough review process, including discussions among reviewers and careful consideration of their detailed feedback, it is my recommendation that this submission should be rejected from ICLR this year. While the paper achieves a outstanding performance on the large-scale datasets, there are several critical issues that undermine its contributions.

1. The proposed method, CDA, is specifically tailored for the baseline method SRe$^2$L. When applied to other dataset distillation methods, CDA yields only marginal performance improvements compared to the significant enhancement observed with SRe$^2$L.

2. The paper lacks technical innovation. The baseline method SRe$^2$L has already brought dataset distillation into the era of large-scale datasets. The proposed method in this paper does not propose constructive improvements at the fundamental algorithmic level.

3. The authors do not provide adequate insights into the derivation of CDA or discuss the intrinsic factors that contribute to performance enhancement. This oversight results in a predominantly heuristic approach in the paper.

The paper does not currently meet the high standards of innovation and clarity expected for ICLR publications. In light of these concerns, I recommend rejection. However, I encourage the authors to address these issues and consider submitting their work to future conferences after substantial revision and improvement.

**Justification For Why Not Higher Score:**

1. The proposed method, CDA, is specifically tailored for the baseline method SRe$^2$L. When applied to other dataset distillation methods, CDA yields only marginal performance improvements compared to the significant enhancement observed with SRe$^2$L.

2. The paper lacks technical innovation. The baseline method SRe$^2$L has already brought dataset distillation into the era of large-scale datasets. The proposed method in this paper does not propose constructive improvements at the fundamental algorithmic level.

3. The authors do not provide adequate insights into the derivation of CDA or discuss the intrinsic factors that contribute to performance enhancement. This oversight results in a predominantly heuristic approach in the paper.

**Justification For Why Not Lower Score:**

NA

---

### Decision · Program_Chairs · 2024-01-16

Reject